# Theoretical uncertainties for global satellite-derived burned area estimates

James Brennan[1,2], Jose L Gómez-Dans[1,2], Mathias Disney[1,2], and Philip Lewis[1,2]

[1]NERC National Centre for Earth Observation, UK
[2]Dept. of Geography, University College London, UK

**Correspondence:** James Brennan (james.brennan.11@ucl.ac.uk)

**Abstract.** Quantitative information on the error properties of global satellite-derived burned area (BA) products is essential for evaluating the quality of these products e.g. against modelled BA estimates. We estimate theoretical uncertainties for three widely-used global satellite-derived BA products using a multiplicative triple collocation error model. The approach provides spatially-unique uncertainties at $1°$ for the MODIS Collection 6 burned area product (MCD64); the MODIS Collection 5.1 MCD45 product and the FireCCI50 product for 2001-2013. The uncertainties on mean global burned area for three products are $3.76 \pm 0.15 \times 10^6$ km$^2$ for MCD64, $3.70 \pm 0.17 \times 10^6$ km$^2$ for FireCCI50, and $3.31 \pm 0.18 \times 10^6$ km$^2$ for MCD45. These correspond to relative uncertainties of 4–5.5% and also indicate previous uncertainty estimates to be underestimated. Relative uncertainties are 8–10% in Africa and Australia for example and larger in regions with less annual burned area. The method provides uncertainties that are likely to be more consistent with modelling and data analysis studies due to their spatially explicit properties. These properties are also intended to allow spatially explicit validation of current burned area products.

## 1   Introduction

Several global satellite-derived burned area (BA) products have been generated for the past two decades. These products generated from coarse spatial resolution (250m–1000m) satellite imagery have provided vital information to fire-related disciplines (Mouillot et al., 2014). They have provided new information on global pyrogeography and changes in fire occurrence (Archibald et al., 2013; Andela et al., 2017); been used to calibrate and validate fire models within dynamic global vegetation models (DGVMs) (Hantson et al., 2016; Thonicke et al., 2001); as well as to drive 'bottom-up' estimates of fire emissions (van der Werf et al., 2017; Seiler and Crutzen, 1980). Despite such value, the true information content of such datasets is still to be fully quantified. The trust that users can place into these products can be improved by providing estimates of product uncertainty. This entails providing a quantitative statement about the lack of knowledge of the true burned area – described by a probability density function (PDF) characterising the range and likelihood of possible values (ISO/BPIM, 2008; IPCC 2006). Burned area products display large intra- and inter-annual differences in the magnitude and timing of fire activity (Giglio et al., 2010; Padilla et al., 2015). Humber et al. (2018) indicated that the range of total recorded burned area for 2005-2011 varied by 90% between four global satellite-derived burned area products. These ranges imply considerable uncertainty in the global burned area satellite record. Previous burned area product intercomparison initiatives have attempted to explore and explain

the spatial and temporal differences observed between different products. Large differences between product estimates have been highlighted in tropical regions, boreal Eurasia and sub-Saharan Africa (Humber et al., 2018; Giglio et al., 2010). These divergences have been interpreted to be driven by differences in the observing properties of the satellites used to create products, as well as the mapping algorithms used within each product. A key determinant on the accuracy of burned area detection originates from the spatial mapping scale of products, with evidence that products produced from higher resolution observations have reduced omission errors (Roy and Boschetti, 2009). Others have highlighted the importance of the temporal revisit time of the utilised satellite instrument (Boschetti et al., 2004). Similarly, the role of persistent cloud cover in some regions has been highlighted, with large divergences between burned area estimates in southeastern Asia ascribed to differences in algorithm observational requirements (Humber et al., 2018). Differences in algorithm decisions and assumptions have also been emphasised, with evidence that even non-vegetated areas (i.e. deserts) display burning for some products (Giglio et al., 2010).

While these intercomparison and validation exercises have provided insight into product performance, the global estimation of product uncertainties from such exercises is difficult. Even the largest and most sophisticated validation datasets correspond to only a small sampling of global fire activity, and it is not clear whether this is sufficient information to build an understanding of uncertainties at global and decadal scales. Uncertainty quantification (UQ) has been requested by users of burned area products for several years (Mouillot et al., 2014; Rabin et al., 2017). Fire modellers have indicated that the discrepancies between products and lack of systematic uncertainty information have restricted efforts for improving models. Poulter et al. (2015) considered the sensitivity of a dynamic global vegetation model to the driving satellite burned area product used. They indicated that the model displayed large sensitivities to deviations between the satellite products and greater UQ would help to drive improvements in model development and benchmarking. Concerns have also been expressed about the calibration of fire models against burned area products which lack the necessary uncertainty information to evaluate model performance in a systematic manner against the observations (Yue et al., 2014; Knorr et al., 2014).

This paper addresses the requirement for uncertainties on global satellite-derived burned area by estimating the uncertainties of three widely-used burned area products. Section 2 outlines the sources of uncertainties in burned area products and previous estimates of uncertainties. Section 3 then describes the uncertainty estimation procedure used here. Section 4 presents the results of the uncertainty model and compares the uncertainty estimates against two other available estimates of burned area uncertainties. Section 5 considers the assumptions of the error model used and Section 6 discusses potential mechanisms for the reported uncertainties. Section 7 concludes the paper.

## 2 Uncertainties in burned area products

### 2.1 Sources of uncertainty

The production of global records of burned area involves the processing of considerable volumes of coarse-resolution satellite observations. Burned area products lie at the top of a measurement process involving the transformation of the initial satellite measurements to higher-level burned area inferences (Merchant et al., 2017). Uncertainties enter this measurement process at

all levels. The initial satellite measurements are not error-free and these uncertainties are thus propagated through the burned area retrieval algorithm. In addition, the detection of changes and the attribution to burning naturally involve an uncertain inference on the state of the land surface.

The optical surface reflectance and thermal measurements used to map burned area have inherent uncertainties due to the measurement process. The optical surface reflectance products, for example, are themselves derived geophysical variables which involve the application of retrieval algorithms (e.g. atmospheric correction), introducing additional uncertainties into the measurement (Vermote et al., 2002).

The sampling provided by Earth-orbiting sensors contributes additional uncertainties. Satellite instruments collect measurements of an area of the land surface infrequently in time and from different acquisition geometries of the Sun and sensor. Variations in sampling geometry alter both the ground area sampled by the sensor and the apparent reflectance signal. The wide-swath instruments typically used to produce burned area products provide the temporal sampling necessary to detect the ephemeral signal of fire on the land surface. However, large variations in the sampling geometries from these sensors complicate the detection of changes in the land surface related to fire (Roy et al., 2005). Zhang et al. (2003) found that changes in the viewing geometries between pre- and post-fire reflectance resulted in enhanced difficulty of identifying burned areas in boreal forests. Similarly, variations in the area sampled lead to a significant proportion of the recorded signal originating from outside of the pixel. Huang et al. (2002) indicated that the blurring due to the sensor PSF reduced the accuracy of land cover classifications by around 5%.

The temporal sampling of the land surface is a key feature in the ability to resolve burned areas. Most significant for burned area mapping is the relationship between observation opportunity and the persistence of the burn signal on the land surface. This persistence is determined by the characteristics of the post-fire recovery of vegetation, as well as the dissipation of ash and char from the burn site. In boreal forests, an observable signal may last many years, savannas typically register a persistent signal for only a few weeks, and the subsequent ploughing of croplands may remove evidence for burning within a week (Sukhinin et al., 2004; Trigg and Flasse, 2000; Hall et al., 2016). The timely observation of the land surface pre- and post-fire then serves as a key determinant on the successful detection of burned areas. Melchiorre and Boschetti (2018) indicated that the median global persistence of an observable burn signal is 29 days, and that within 48 days 87% of global burned area is undetectable.

The procedures and assumptions built into detection algorithms also determine the error properties of individual products. Burned area products display regional disparities in performance that are in line with differences in fire characteristics (Padilla et al., 2015). Developers of burned area products have previously highlighted limitations within their algorithms. Simon et al. (2004) indicated that parameters within their algorithm may lead to commission/omission errors in different regions. Roy et al. (2005) suggested that their algorithm may miss fires which display rises in post-fire reflectance. And Giglio et al. (2009) suggested that the assumption of a decline in a post-fire vegetation index within their algorithm is not met in around 20% of fires over validation data from north-western Australia.

## 2.2 Present uncertainty estimates

Previous estimates of product uncertainties have been largely driven by validation initiatives. In these analyses, product commission and omission errors have been computed in comparison to reference datasets, which are typically generated by the manual or semi-automated mapping of area burned from higher resolution images. The extents of these validation exercises range from regional comparisons against a few selected sites to larger global validation designs (Roy and Boschetti, 2009; Boschetti et al., 2016; Padilla et al., 2017, 2015). The derived validation statistics are then interpreted as providing estimates of the uncertainties of the product in light of these commission/omission statistics. The clearest example of this is the estimate of burned area standard error $\sigma_A$ provided in the Global Fire Emissions Database (GFED) 4 product (Giglio et al., 2010):

$$\sigma_A^2 = c_B A \tag{1}$$

where $A$ is the aggregated burned area in the grid cell. $c_B$ serves as an uncertainty coefficient which scales the standard error based on an analysis of residuals against Landsat validated burned area.

A natural concern that arises out of these approaches is the quality of the sampling provided by such validation datasets. Even larger and more systematic validation efforts may still provide only a limited sampling of the true uncertainties. For example, the validation of products against 103 validation sites by Padilla et al. (2015) are derived from active fire observations, which display their own issues and uncertainties (Giglio et al., 2006a). Similarly, the challenge of generating sufficient validation data to enumerate global uncertainties in burned area is considerable. The estimated uncertainties provided by GFED4 are derived from three unique values for $c_B$ (covering Siberia, Southern Africa and the western United States), and regions not sharing sufficient similarities with these are given a median value of $c_B$ (Giglio et al., 2010). An additional limitation of the regional enumeration of $c_B$ is that it must replicate contributions from additional uncertainty sources. These will be features such as variations in cloud cover obscuring burned area detection, and uncertainties arising from variations in the distribution and local mixture of vegetation type. This variability will alter the value of $c_B$ within each region.

An exception to this approach is provided by the FireCCI version 5.0 product (FireCCI50) which provides per-pixel estimates of uncertainty in the detection of burned areas (Chuvieco et al., 2018). These uncertainties are computed by considering a number of features of the detection problem such as the number of observations available, and the magnitude of the reflectance change signal. These pixel level uncertainties are then aggregated into the lower resolution FireCCI50 product to provide per 0.25° grid cell standard errors. The validity of these standard errors will be dependent upon the quality of the per-pixel uncertainty estimates (in terms of modelling the true uncertainty) and the aggregation process from pixel to coarser grid cell scales (Bellprat et al., 2017).

In the absence of product provided uncertainty estimates, others have also derived estimates of uncertainties. Le Page et al. (2015) proposed uncertainties of 25-50% in burned area as provided by GFED4 based on an inspection of the GFED data. Most frequently the range in burned area reported by different products has been used to provide upper and lower bounds on global burned area (Rabin et al., 2017; Forkel et al., 2019; Poulter et al., 2015; Knorr et al., 2012). The large uncertainty in global burned area implied by this figure contributes considerably to emissions uncertainties (Knorr et al., 2012). It also introduces

additional problems into the evaluation of the performance of fire models against satellite-derived observations (Rabin et al., 2017).

## 3 Materials & Methods

### 3.1 Burned area datasets

The present study estimates theoretical uncertainties for three global burned area products. The Moderate Resolution Imaging Spectroradiometer (MODIS) Collection 6 burned area product (MCD64C6) provides a global record of burned area for the MODIS period (i.e. 2000–present). The algorithm uses active fire observations to refine a classifier based on the application of a temporal change spectral index derived from MODIS short-wave infrared channels 5 (1230–1250nm) and 7 (2105–2155nm) (Giglio et al., 2018).

The MODIS Collection 5.1 burned area product (MCD45C5.1) was produced with a different algorithm and provides a global record of burned area for a reduced period covering 2000-2016. The product uses a multi-temporal modelling algorithm which flags for changes in the land surface based on differences between predicted and observed reflectance. The algorithm then filters changes to those that match the expected reflectance characteristics of burned surfaces in the near-infrared (841–876nm) and short-wave infrared (1230–1250nm). The algorithm does not utilise active fire observations (Roy et al., 2005).

The ESA Climate Change Initiative Fire product (FireCCI50) provides global burned area for 2001-2016. The algorithm uses changes in MODIS near-infrared (841–876nm) surface reflectance inside a classifier that like MCD64C6, is locally trained with active fire observations from the MODIS sensors (Chuvieco et al., 2018). The product is novel in that it provides burned area at a spatial resolution of 250m compared to the 500m spatial resolution of the other two products. This limits the algorithm to use only the red and near-infrared spectral bands.

The MCD45C5.1 product has now been deprecated by the Collection 6 MCD64 algorithm. The operational 1km Copernicus burned area product was also considered however issues have been found in the product which has resulted in the product being withdrawn for re-processing (Service). The newer 300m Copernicus burned area product covers a more limited temporal span from 2014–present. In terms of data set selection the three chosen products represent the longest available combined satellite record.

### 3.2 Computation of uncertainties

Stoffelen (1998) first proposed triple collocation (TC) as a method to estimate uncertainties in three collocated data products. The method has now been used across a considerable range of remote sensing derived geophysical variables including soil moisture, precipitation, leaf area index and fraction of photosynthetically absorbed radiation (Gruber et al., 2016; Roebeling et al., 2012; Fang et al., 2012; D'Odorico et al., 2014). Consider three observational records $X_1, X_2, X_3$ of a variable with an 30 unknown but true value $T$. The TC error model specifies that each observational record may be related to the truth via a linear

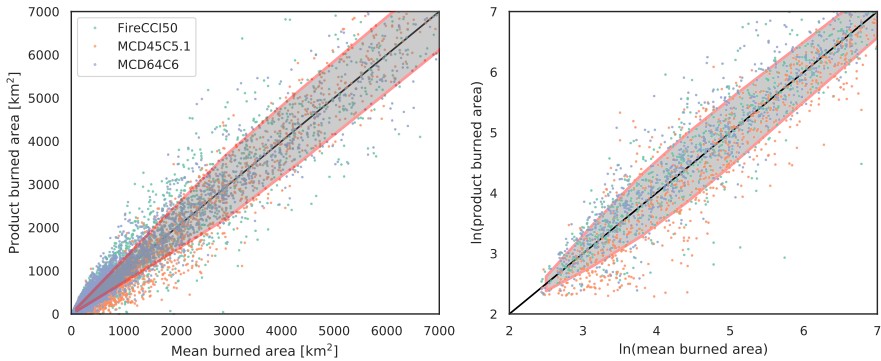

**Figure 1.** Differences between the burned area reported by the three products and the mean of the three products. Also shown is the standard deviation of the products (grey) binned by the mean burned area of the three products. Increasing standard deviation with the magnitude of burned area implies heteroscedastic errors , while log-transformed burned area have errors which are more homoscedastic.

measurement equation:

$$X_1 = \alpha_1 + \beta_1 T + \epsilon_1 \tag{2}$$

$$X_2 = \alpha_2 + \beta_2 T + \epsilon_2 \tag{3}$$

$$X_3 = \alpha_3 + \beta_3 T + \epsilon_3 \tag{4}$$

where $\alpha$ and $\beta$ represent additive and multiplicative biases respectively. $\epsilon$ denotes the residual (random) errors of the relation and are considered here to be normally distributed.

     As posited, the three measurement equations indicate a system that is under-determined. However by making three assumptions, the system can be solved to provided estimates of the random errors of each product. First, each product is assumed to have zero mean residual errors ($E[\epsilon] = 0$). Second, the errors of each product are assumed to be uncorrelated (but not necessar-

ily independent) with each other. Finally, the random error distribution is assumed to be uncorrelated with the true value $T$, as systematic errors are incorporated into $\beta$. The last assumption is not met for geophysical variables which show random errors that are functionally related to the magnitude of the signal (Tian et al., 2013).

     Figure 1 shows mean annual burned area of the three products against individual product estimates. The shaded area represents the standard deviation between the products binned by the mean of the three products. It can be observed that the

deviations between the products grow with the magnitude of burned area reported. This indicates that the constraint imposed on the burned area becomes more uncertain with the magnitude of burned area detected. This occurs because the random errors in burned area are *heteroscedastic* (Giglio et al., 2006b). The TC model in eq. 2-4 assumes however that the random errors $\epsilon$ are homoscedastic – in that the error variance model $\epsilon = \mathcal{N}(0, \sigma^2)$ is not a function of the true (unobserved) burned area. This feature of the errors is common to several other geophysical variables (e.g. precipitation, above-ground biomass) (Tian et al.,

2013; Alemohammad et al., 2015; Gonzalez de Tanago et al., 2018).

In log space however the differences between products do not increase with the logarithmic burned area and are closer to being homoscedastic. Alemohammad et al. (2015) proposed that for heteroscedastic datasets, an alternative TC error model is suitable in which the random error is a multiplicative signal on the truth $T$. Instead, the error model for $X$ can be related as:

$$X = \alpha T^\beta e^\epsilon \tag{5}$$

where $\alpha$ is a multiplicative error, $\beta$ is the deformation error and $e^\epsilon$ is the residual (random error). Taking the natural logarithm of equation 5 leads to an additive measurement model:

$$ln(X) = \alpha + \beta ln(T) + \epsilon \tag{6}$$

with the assumption that in the log-space, the random errors are normally distributed $\epsilon = \mathcal{N}(0, \sigma^2)$. Representing $x = ln(X)$ and $t = ln(T)$, eq. 6 is equivalent to:

$$x = \alpha + \beta t + \epsilon \tag{7}$$

which provides a linear system equivalent to eq. 2. Given the same assumptions of the classical TC method, the residual error estimates of each product (in log-space) can be derived from the following manipulations of the sample covariance matrix $\mathbf{C}$ of the three log-transformed products (McColl et al., 2014):

$$\sigma_1^2 = \mathbf{C}_{11} - \frac{\mathbf{C}_{12}\mathbf{C}_{13}}{\mathbf{C}_{23}} \tag{8}$$

$$\sigma_2^2 = \mathbf{C}_{22} - \frac{\mathbf{C}_{12}\mathbf{C}_{23}}{\mathbf{C}_{13}} \tag{9}$$

$$\sigma_3^2 = \mathbf{C}_{33} - \frac{\mathbf{C}_{13}\mathbf{C}_{23}}{\mathbf{C}_{12}} \tag{10}$$

A requirement of the TC method is that the three datasets explicitly cover the same temporal and spatial domain and are of the same variable (Yilmaz and Crow, 2014). To achieve this, the three burned area datasets were aggregated to a shared temporal and spatial grid. The three products were aggregated from the original pixel resolution products to a common sinusoidal grid $g$ with a resolution of $1°$ at the equator. For each 16-day period between January 2001 – December 2013, the burned area reported by each product within the cell $g(t, x, y)$ was aggregated to form a full temporal record for each cell through time of length $N_t$. The temporal span of the datasets provided potentially $N_t = 286$ observations. A feature of solving the multiplicative error model in log-space is that any product that reports no burned area will prevent the estimation of the covariance matrix $\mathbf{C}$. As a result, any 16-day period where at least one product reported no reported burned area was excluded. This meant that approximately 40% of cells globally had no agreed burned area between the products, and therefore do not have error estimates. Nevertheless, the major fire regions are well sampled across the record (see figure 2). The TC method is able to sample the majority of the reported fire activity by the products. Total burned area over the study period for cells which do not have associated uncertainties is less than 0.5% of the total burned area of each product.

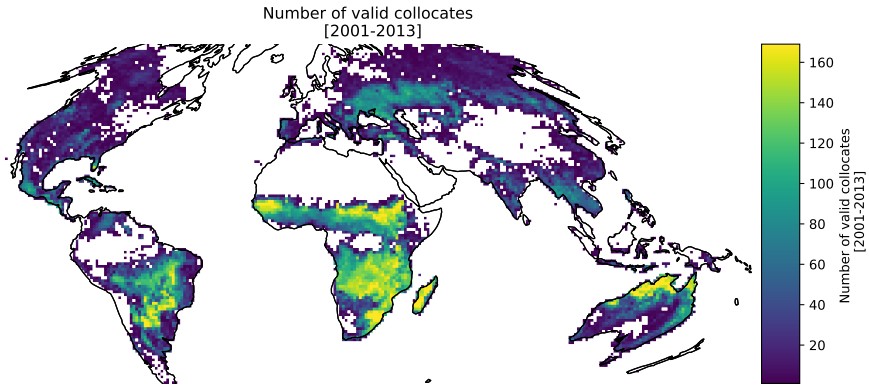

**Figure 2.** Number of valid collocates for 2001-2013.

### 3.2.1 Annualised uncertainties

Beyond product standard errors, annualised uncertainties on the total burned area are also of particular interest to the users of burned area products. To produce 16-day uncertainties in the burned area for each product, reconsider the error model specified in eq. 5. The random errors back-transformed into burned area are defined by a log-normal distribution specified by

5 Log-normal($\mu = 0, \sigma^2$). Therefore the distribution of 16-day burned area $P(X)$ can be defined in reference to eq. 5 as

$$P(X) = X_o e^{\mu + \sigma Z}, \tag{11}$$

where $X_o$ is the observed burned area for the product and $Z$ the standard normal distribution. To produce an annual uncertainty estimate, each 16-day burned area distribution $P(X)$ was sampled from and integrated over the year to provide a distribution of annual burned area for each grid cell. The independence assumption of individual observation errors in this scheme is also

10 a requirement of the TC method (Gruber et al., 2016). To summarise the annual distribution, it was then approximated as a normal distribution based on matching the moments of the samples. Figure 3 shows an example of the procedure for producing 16-day and annualised uncertainties for an area covering Northern Australia. Large absolute uncertainties are associated with the peak in the burning season here.

Given the regional variability in absolute burned area, the relative magnitude of the annual uncertainties to the reported burned area of each product was also considered. The relative uncertainty in mean annual burned area is defined by:

$$\text{rel. unc.}\% = 100 \times \frac{\sigma_{\text{year}}}{\text{BA}_{\text{year}}} \tag{12}$$

where $\text{BA}_{\text{year}}$ is the total burned area reported by the product for the grid cell for each individual year.

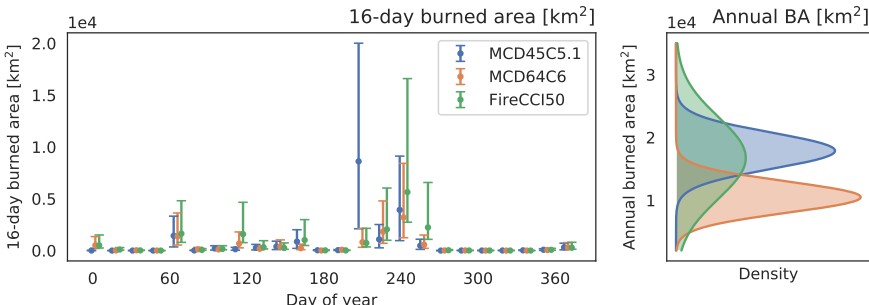

**Figure 3.** 16-day and annual uncertainties for a grid cell covering Northern Australia. Left) The multiplicative error model provides unique uncertainties on each 16-day observation for each product (95% confidence intervals shown). Right) To produce an annual uncertainty on the reported burned area, these are aggregated to produce an annual distribution which is then approximated as a normal distribution.

### 3.2.2 Regional and global uncertainties

Given that we may expect the performance of each product to vary with the local fire behaviour, we considered the uncertainty estimates with regard to the IGBP Land Cover Type Classification provided in the MODIS Collection 6 land cover product (MCD12Q1.006) (Friedl et al., 2010). We simplified the University of Maryland (UMD) land cover classification into five more

primary categories of 1) forest including all forest types, 2) croplands, 3) shrublands including both open and closed shrublands, 3) savannas, and 5) grasslands. The simplified land cover product was then aggregated to the sinusoidal 1° resolution grid by considering the dominant land cover type in each cell. We also considered product errors within the 14 fire regions specified by GFED which have been previously used for regional comparisons of burned area products (Giglio et al., 2013)

     A complicating feature of the aggregation to the regional scales is that the spatial correlation of the uncertainties at the

grid cell level is unknown. It would generally be expected that the uncertainties in adjacent grid cells may be similar, due to correlations in the driving features of the uncertainties e.g. land cover, cloud statistics and algorithmic limitations. The integration of grid cell level uncertainties via an independent quadrature summation would imply a strong constraint on there being no spatial correlation in the uncertainties (Bellprat et al., 2017). Instead, to produce the regional estimates, 16-day burned area for each product was aggregated for the whole region or land cover stratification and the TC error model then applied.

This allows for the effective spatial error correlation in the products to be present in the regional uncertainties while requiring no additional assumptions about the error structure.

### 4 Results

Figure 4 displays global maps of the residual errors (in log-space) for each product. Spatial patterns in uncertainties show general similarities at broad scales. The patterns are also different from the spatial distribution of burning, indicating that

systematic errors are not leaking into the random errors. The largest random errors for each product are located in Eastern

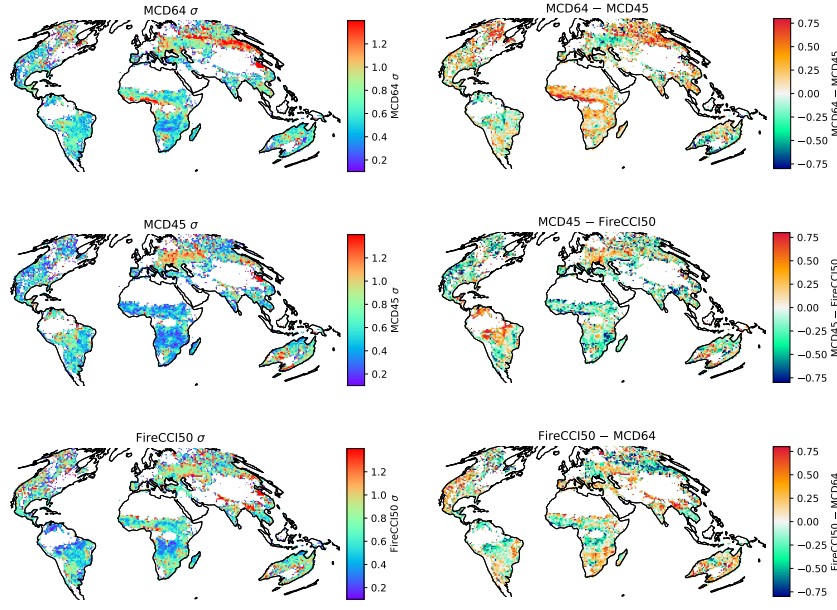

**Figure 4.** Left) TC random errors for the three products and right) differences between product random errors.

China corresponding to regions of agricultural fires. Here errors are greater than 1 for all products, which indicates a random error of greater than 100% in the detected burned area. This would indicate that the level of agreement between the products is lower than the precision of the products.

Local patterns of the errors then diverge for each product. MCD45 has larger random errors in central and eastern Europe, in regions with predominantly agricultural fires. The lowest uncertainties are found in savanna ecosystems of southern hemisphere Africa and northern Australia. MCD64 shows the largest uncertainties in agricultural and tundra regions of eastern Eurasia. It also has the largest uncertainties in Western Africa, in areas where deforestation fires are common. MCD64 has larger uncertainties in savannas relative to MCD45 and lower random errors in areas with agricultural burning. FireCCI has smaller errors in agricultural regions of eastern Eurasia compared to the other two products. FireCCI also has smaller random errors in regions of agricultural burning and deforestation areas around the Amazon compared to MCD45 and MCD64.

Figure 5 displays global maps of mean annual burned area and associated uncertainties for the three products. Between the products, similar spatial distributions in burned area and TC-estimated uncertainties can be observed. The heteroscedastic nature of burned area uncertainties is apparent with standard uncertainties scaling with the magnitude of burned area. Absolute uncertainties for each product are largest in sub-Saharan Africa and northern Australia which corresponds to regions with the greatest burned area. Greater disagreement in the magnitude of burning occurs in regions with less frequent burning or typically compounding factors on detection. In equatorial Asia, MCD64 and FireCCI50 detect respectively 1310% and 940% more burned area than MCD45. Greater detection by MCD64 here has been associated with the use of active fires (Humber et al., 2018). These higher estimates are also better constrained with relative uncertainties of 35% and 36% respectively for

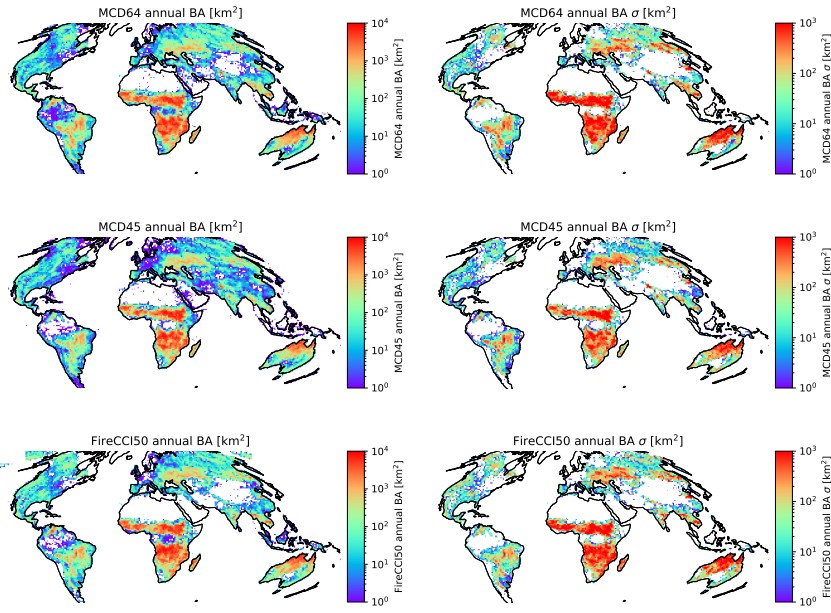

**Figure 5.** Left) Mean annual burned area $\mathrm{km}^2$ and right) associated standard errors of mean annual burned area $\mathrm{km}^2$ .

FireCCI50 and MCD64; compared to a higher relative uncertainty of 70% on the MCD45 burned area. FireCCI50 detects 66% more burned area in the agricultural burning regions of central and eastern Europe than MCD64 and 48% more than MCD45. However, the large uncertainties on these estimates indicate them to be consistent within the uncertainties: with relative uncertainties of 141% on FireCCI50, 168% on MCD45 and 95% on MCD64. Regions where MCD45 reports no

burning prevents the estimation of TC-uncertainties due to the requirement of the multiplicative error model used here. This is most noticeable in equatorial Asia and South America.

Globally, MCD64 reports the greatest mean annual burned area $3.76 \pm 0.15 \times 10^6 \mathrm{\ km}^2$. This is followed by FireCCI50 which reports $3.70 \pm 0.17 \times 10^6 \mathrm{\ km}^2$ and MCD45 $3.31 \pm 0.18 \times 10^6 \mathrm{\ km}^2$. In terms of relative uncertainties, MCD64 has the smallest relative uncertainty of 3.9%, FireCCI50 4.5% and MCD45 has the largest (5.5%). MCD64 and MCD45 provide consistent

estimates of mean annual burned area for 76% of grid-cells with TC-estimated uncertainties. In these locations, estimates from both products are within the range of standard uncertainties provided from the TC method. MCD64 and FireCCI50 agree across a slightly broader spatial extent, with 80% of available cells agreeing within the uncertainties of each product. MCD45 and FireCCI50 have the lowest agreement of the three products, with consistent estimates across 72% of TC-cells. Figure 6, shows locations where all three products agree within their standard uncertainties for mean annual burned area. Overall, all

three products agree within their uncertainties for 60% of available TC cells. Within a broader distribution of two standard errors, the three products agree across 85% of the valid cells. Regions where the products do not agree within two standard deviations are concentrated in equatorial Asia, the northern Amazon region, the south-western United States, and parts of the Indian subcontinent.

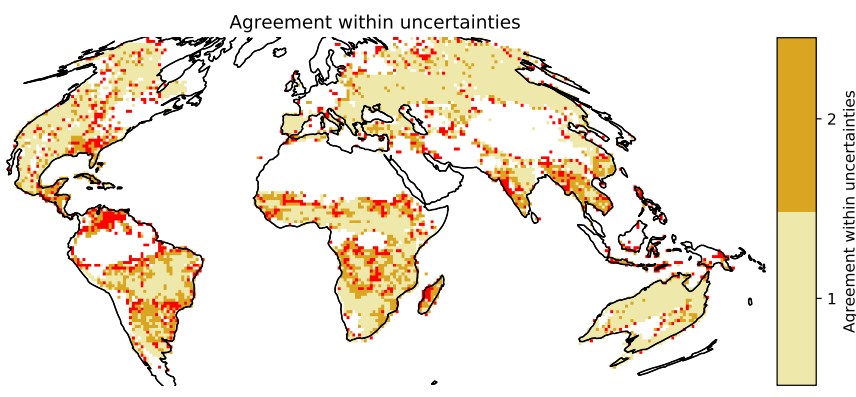

**Figure 6.** Consistency of mean annual burned area for the three products. Light brown regions correspond to regions where all three products agree within one standard error. Brown regions correspond to agreement within two standard errors. Red regions indicate areas which do not agree within two standard errors.

Figure 7 shows a regional breakdown of mean annual burned area and uncertainties stratified by land cover. Globally, burned area estimates are most uncertain for cropland and shrublands for all products. All three products perform comparatively better in savannas and grasslands and less well in forested biomes. For nearly all land covers, MCD45 has the largest relative uncertainties of the three products. It has the largest uncertainties in shrublands, with a relative uncertainty of 25%, followed by FireCCI50 (13%) and then MCD64 (8%). The uncertainty for the MCD45 product in shrublands is contributed to in large part by a poor constraint on burning in Australian (AUST) shrublands where the relative uncertainty exceeds 40% ($1.29 \pm 0.56 \times 10^5$ km$^2$), compared to 15% and 8% for the FireCCI50 and MCD64 products respectively. FireCCI50 uncertainties in shrublands are driven by large uncertainties on comparatively small reported shrubland burned area in Central America (CEAM) 765 $\pm$ 1846 km$^2$ and Temperate North America (TENA) 1175 $\pm$ 2115 km$^2$. This contrasts with much smaller uncertainties on a similar reported burned area from MCD64 in Temperate North America (TENA) 1172 $\pm$ 449 km$^2$.

All products have a poor constraint on global cropland burning with relative uncertainties of 8-10%. MCD45 generally has the largest relative uncertainties on cropland burning across all fire regions, with confidence intervals larger than the magnitude of reported burned area for Europe (EURO), boreal Eurasia (BOAS), and equatorial Asia (EQAS). Exceptions are found in temperate North America (TENA) and southeast Asia (SEAS) where MCD45 reports the most cropland burning and also has the lowest relative uncertainties.

An interesting feature occurs in boreal forest ecosystems, where MCD45 and FireCCI have smaller uncertainties in boreal Eurasian (BOAS) forests compared to boreal North American (BONA) forests. Uncertainties for MCD45 are around two times larger in BONA forests, and 40% larger for FireCCI50 in BOAS as compared to BONA forests. Alternatively, MCD64 has lower relative uncertainties in BONA compared to BOAS, with uncertainties 70% larger in boreal Eurasia.

**Table 1.** Mean annual burned area [$\times 10^3$ km$^2$], standard uncertainty [$\times 10^3$ km$^2$] and relative uncertainty [%] for the products by fire region.

| product region | Burned area [$\times 10^3$ km$^2$] | | | Standard uncertainty [$\times 10^3$ km$^2$] | | | relative uncertainty [%] | | |
| --- | --- | --- | --- | --- | --- | --- | --- | --- | --- |
| | FireCCI50 | MCD45 | MCD64 | FireCCI50 | MCD45 | MCD64 | FireCCI50 | MCD45 | MCD64 |
| AUST | 514.39 | 394.02 | 476.27 | 37.19 | 73.84 | 33.43 | 7.23 | 18.74 | 7.02 |
| BOAS | 94.47 | 64.70 | 86.26 | 53.96 | 20.87 | 66.34 | 57.12 | 32.26 | 76.90 |
| BONA | 22.37 | 14.63 | 20.61 | 11.97 | 9.85 | 13.57 | 53.53 | 67.33 | 65.87 |
| CEAM | 22.75 | 13.75 | 25.68 | 9.43 | 8.25 | 5.95 | 41.47 | 60.02 | 23.16 |
| CEAS | 209.51 | 190.49 | 194.89 | 53.58 | 42.11 | 30.49 | 25.58 | 22.10 | 15.64 |
| EQAS | 9.18 | 0.88 | 12.47 | 3.22 | 0.61 | 4.52 | 35.08 | 69.25 | 36.29 |
| EURO | 13.92 | 11.63 | 10.52 | 11.87 | 13.23 | 4.94 | 85.31 | 113.81 | 46.94 |
| MIDE | 10.01 | 16.22 | 12.55 | 5.70 | 9.57 | 0.95 | 57.00 | 59.03 | 7.56 |
| NHAF | 987.23 | 1077.43 | 1032.15 | 266.58 | 126.32 | 188.68 | 27.00 | 11.72 | 18.28 |
| NHSA | 51.68 | 10.50 | 45.14 | 7.27 | 12.66 | 10.83 | 14.07 | 120.65 | 23.99 |
| SEAS | 119.73 | 91.29 | 117.40 | 59.26 | 38.28 | 67.35 | 49.49 | 41.93 | 57.37 |
| SHAF | 1397.68 | 1227.17 | 1413.56 | 103.21 | 145.44 | 167.46 | 7.38 | 11.85 | 11.85 |
| SHSA | 215.15 | 169.19 | 279.61 | 5.44 | 48.83 | 42.77 | 2.53 | 28.86 | 15.29 |
| TENA | 31.68 | 25.77 | 24.42 | 4.05 | 4.45 | 3.39 | 12.78 | 17.26 | 13.90 |
| WORLD | 3701.72 | 3309.44 | 3755.80 | 165.55 | 183.38 | 146.05 | 4.47 | 5.54 | 3.89 |

In the key burning regions of northern Hemisphere (NHAF) and southern hemisphere Africa (SHAF), MCD45 typically has the most constrained estimate of burned area. The three products provide consistent estimates in grasslands and savannas in both regions, with reported burned area for each product agreeing within the uncertainties estimated for all products. The uncertainties are still considerable, however, with relative uncertainties for all three products largest in savannas and grasslands. In these land covers, relative uncertainties exceed 13% in NHAF and 8% in SHAF. This leads to broad standard errors on each product in NHAF, with reported mean annual burned area of $1.03 \pm 0.19 \times 10^6$ km$^2$ for MCD64, $1.07 \pm 0.13 \times 10^6$ km$^2$ for MCD45, and $0.99 \pm 0.27 \times 10^6$ km$^2$ for FireCCI50. Table 1 summarises the mean annual burned area and uncertainties by fire region.

## 4.1 Comparison against other uncertainty estimates

### 4.1.1 GFED4 uncertainties

We contrast the uncertainties from the TC method with two other available uncertainty estimates. First in relation to the MCD64 product we consider the uncertainties provided with the GFED4 burned area product. The GFED4 burned area and uncertainty are derived exclusively from the MCD64 product for the period considered here. GFED4, however, utilised the older MCD64

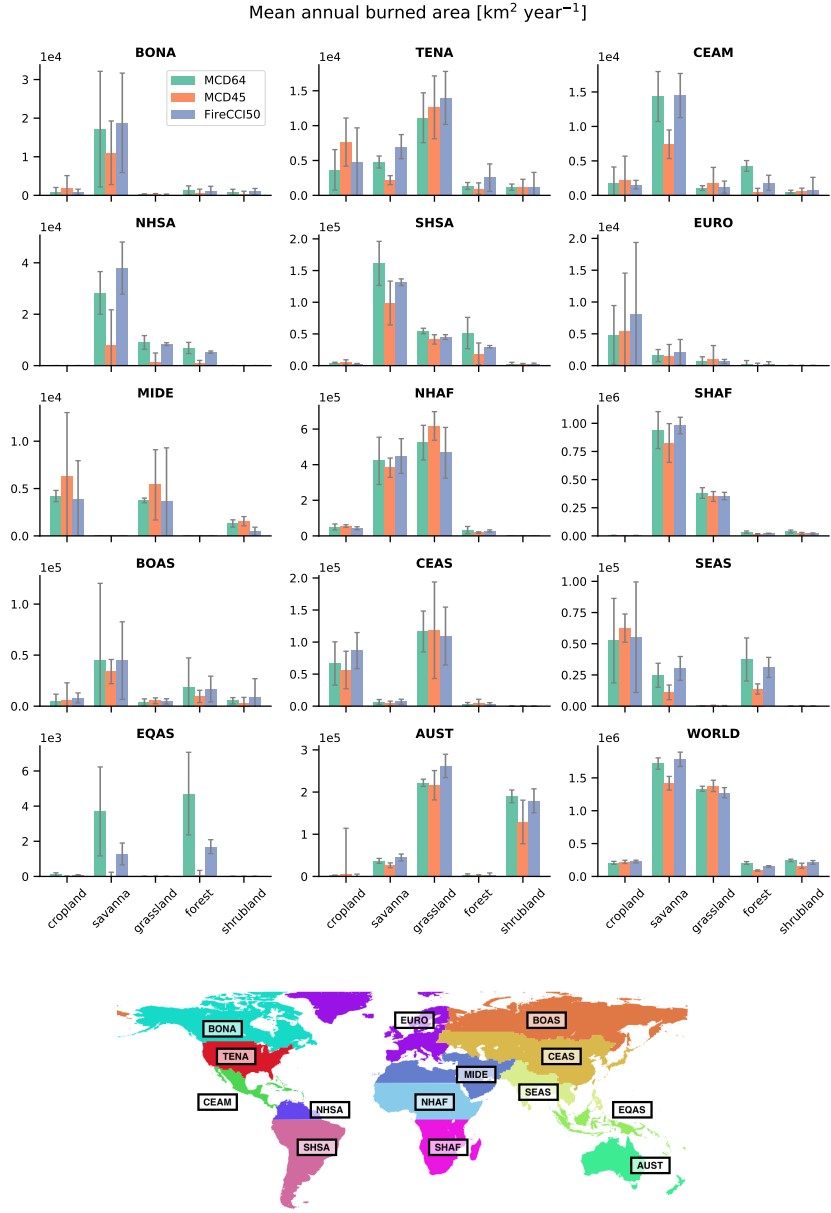

**Figure 7.** Mean annual burned area and uncertainties $km^2/year$ for the fire regions stratified by land cover. BONA) Boreal North America, TENA) Temperate North America, CEAM) Central America, NHSA) Northern Hemisphere South America, SHSA) Southern Hemisphere South America, EURO) Europe, MIDE) Middle East, NHAF) Northern Hemisphere Africa, SHAF) Southern Hemisphere Africa, BOAS) Boreal Asia, CEAS) Central Asia, SEAS) Southeast Asia, EQAS) Equatorial Asia, AUST) Australia & New Zealand.

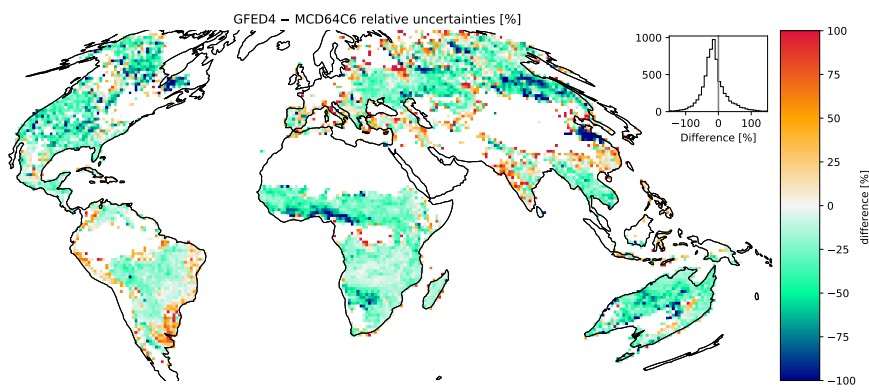

**Figure 8.** Differences in relative uncertainties between GFED4 and TC-estimated relative uncertainties.

Collection 5.1 product, which detects significantly less global burned area than the present Collection 6 product (Giglio et al., 2018). Nevertheless, in the absence of other uncertainty estimates, it is sensible to consider the relative uncertainties for the GFED4 product against the TC estimates. To align the uncertainties with those provided by the TC method, the total annual burned area uncertainties were considered. To produce annual uncertainties for GFED4, the monthly variances provided by the GFED4 product were added in quadrature.

Figure 8 shows global differences between mean annual relative uncertainties in GFED4 vs TC derived uncertainties. TC uncertainties generally exceed GFED uncertainties in most regions. The global median for TC uncertainties is 38% and GFED 34%; however mean global GFED uncertainties exceed those provided by the TC method. Mean global GFED uncertainties are 65% compared to 45% provided by the TC method, though this figure is skewed by a greater range in the GFED uncertainties (GFED interquartile range (IQR): 15% – 80% vs TC IQR: 26% – 57%). Areas of higher TC uncertainties are found in the agricultural burning regions of northern China and eastern Russia, where TC uncertainties exceed GFED by 70-100%. TC uncertainties also exceed GFED uncertainties in western Africa (90%) and areas of North America, especially in boreal forest regions of eastern Canada. GFED uncertainties also exceed TC uncertainties in several regions. For example, GFED uncertainties are larger in boreal Eurasia (40-60%), eastern India (30-70%) and parts of South America (35-65%).

We conceive two probable causes for differences between the two uncertainty estimates. Primarily, GFED4 is based on an older collection of the MCD64 product which detected globally around 26% less burned area than the present Collection 6 product (Giglio et al., 2018). An equally important consideration is that the uncertainty assumptions of the two methods are different. For the GFED uncertainties, Giglio et al. (2010) indicated that these are likely to be conservative due to the potential cancelling of omission and commission errors in the total reported burned area, with the effect being that GFED uncertainties are also likely over-estimated for the MCD64 Collection 5.1 product. The TC method accounts for any potential cancelling of errors by focusing on the observed burned area irrespective of the error source.

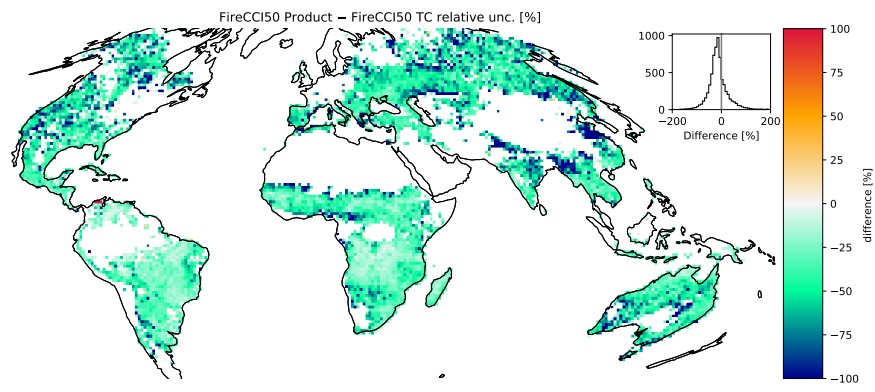

**Figure 9.** Differences in relative uncertainties between product uncertainties for FireCCI50 and TC-estimated relative uncertainties.

### 4.1.2 FireCCI50 product uncertainties

The FireCCI50 Climate Model Grid (CMG) product also provides standard errors per grid cell at the coarse spatial resolutions considered here. These are produced from an aggregation of individual uncertainties in the 250m pixel product to produce fortnightly standard errors in burned area. In the same manner as with the GFED4 uncertainties, we produce annual uncertainties
from the FireCCI50 product by adding the uncertainties in quadrature for each fortnightly product.

The uncertainties provided with the FireCCI50 product represent the first attempt to provide a full uncertainty traceability chain for burned area datasets. We find that the reported uncertainties are considerably smaller than those provided by the TC error model as well as the uncertainty estimates provided by GFED4. Figure 9 shows a comparison of relative uncertainties for TC-derived uncertainties and the uncertainties provided with the FireCCI50 product. TC uncertainties exceed product
uncertainties in 98% of the valid grid cells. Globally, the median relative uncertainty implied by the product is 2% compared to 41% from the TC uncertainties. The product uncertainties have a much smaller global range (IQR: 1–5%) compared to the TC estimate (IQR: 27% – 58%). The difference between TC uncertainties and product uncertainties are largest in cropland areas of northern China (150-200%), eastern Russia (50-100%) and eastern India (60-120%). TC uncertainties are also around (70-100%) larger in regions of the western United States.
Figure 10 shows an example of the pixel level uncertainties provided with the FireCCI50 product. Reference burned area is overlaid from the analysis of two Landsat acquisitions. We see that the product correctly detects the larger burn scars in the image extent. For these larger burn scars, the provided confidence is 70-100%. However, smaller burn scars which are not classified as burned by the algorithm show burn probabilities which are similar to the unburned background (20-40%). These values do not correspond well with the likely fire signal at these locations, with the apparent pattern in unburned confidence
values arising from the interpretation of the composited observations used within the algorithm.

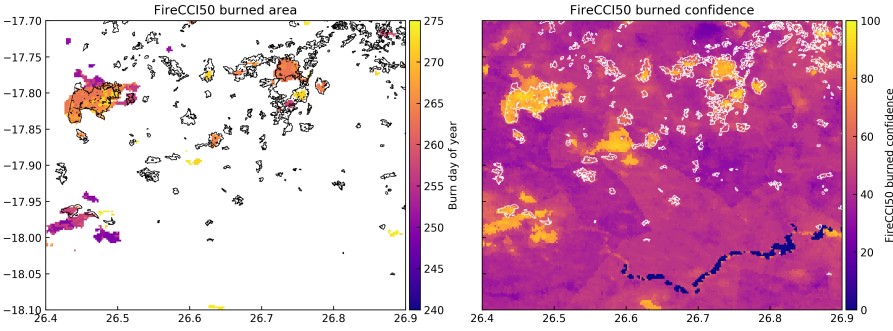

**Figure 10.** Example of the pixel level uncertainties (burned confidence) provided with the FireCCI50 product. The area covers northern Zimbabwe for the period September 2008. Landsat derived burned area is overlaid.

## 5 Considerations of the TC error model

As previously indicated in section 3.2, the TC error model has several key assumptions which must be considered. An initial requirement of the TC method is that the three products correspond to three temporally and spatially collocated data products. Here, this was achieved by considering the products at coarse spatial and temporal scales. The aggregation of daily pixel products to 16-day windows should help to reduce the influence of differences in reporting dates of fires between products. Similarly, the aggregation to a $1°$ spatial resolution grid reduces the chance of highly local differences in reported burned area and therefore should provide more robust estimates for each product. Nevertheless, due to the requirements of the TC method, around 40% of global land cells do not have uncertainties – although this figure includes desert regions. Zwieback et al. (2012) indicated that the relative error in uncertainty estimates from the TC method can be approximated by $\sqrt{\frac{5}{n}}$, where $n$ is the valid number of collocated observations used to compute the product covariance matrix. Users should be aware that the accuracy of uncertainties in regions with less frequent burning will therefore be lower than those regions with longer fire seasons. Given the available temporal span of the products, the mean global relative uncertainty in TC error estimates is expected to be around 33%.

The most significant assumption of the TC method for the presented analysis is that the products do not have error cross-correlations (ECC) (Zwieback et al., 2012; Gruber et al., 2016). ECC structures between burned area products may occur due to 1) the use of the same satellite instruments, 2) shared observation opportunity at the $1°$ spatial scale and 3) similarities in the retrieval algorithms. We now consider each. A key concern is that the three products all utilise observations from the MODIS instruments. All three products utilise MODIS surface reflectance measurements; with FireCCI50 and MCD64 additionally using MODIS active fire detections. In terms of the second ECC source, grid cell uncertainty estimates may also be affected by the general observational opportunity available within the TC cell. Active fire products have a better sampling at higher latitudes relative to the equator (Giglio et al., 2006b), and persistent cloudiness may introduce additional error correlations between the products. Finally, similarities within the mapping algorithms may introduce additional ECC sources. For example, similar thresholds on fire-related changes in reflectance may cause error correlations between the products. In regards to each

source of potential ECCs, we judge that product uncertainties are most significantly determined by algorithmic decisions. This is because the three algorithms use considerably different decision structures for mapping the pixel level burned areas. For example, while MCD64 and FireCCI41 both use active fire observations, the two algorithms utilise distinct expectations of fire properties in different spectral regions. Similarly, several intercomparison activities of these three products have indicated considerable differences between estimates at both the pixel level product and regional burned area estimates(Humber et al., 2018; Padilla et al., 2015).

We also stress that the uncertainties estimated with the TC method likely represent a lower bound on the true uncertainties of these products. The TC measurement model can only explicit estimate random errors but not systematic errors (i.e. bias) present in the data products from fires which are undetectable. The under-estimation bias observed for these coarse-resolution products in validation studies indicates that the products likely have considerable systematic errors. Chuvieco et al. (2018) have estimated that the FireCCI50 product has global omission errors of 70% and MCD64C6 62%, which are partially balanced by commission errors of 50% and 35% respectively. Roteta et al. (2019) also indicated that a higher spatial resolution 20m burned area product provided 80% more burned area than the MCD64C6 product for sub-Saharan Africa, which while not providing a true validation indicates considerable biases in coarse-resolution products. Users should be aware therefore that the likely systematic biases in coarse resolution products mean that the TC uncertainties provide a lower bound on the true uncertainty.

## 6 Discussion

This study has estimated theoretical uncertainties for three global satellite-derived burned area datasets. This study provides an update on ongoing efforts to provide quantitative uncertainties for remotely sensed global burned area estimates initiated with GFED4 (Giglio et al., 2006b) and continued within the FireCCI products (Chuvieco et al., 2018). Within the four-stage validation scheme developed for land remote sensing products developed by the CEOS Land Product Validation (LPV) group, the majority of current burned area products have only achieved stage three validation (Boschetti et al., 2009; Morisette et al., 2006; Chuvieco et al., 2018; Boschetti et al., 2016; Padilla et al., 2017). Meeting the stage four requirement for statistically robust and validated uncertainties remains an open challenge for the burned area community. While new large scale validation datasets of burned area have been recently developed (Chuvieco et al., 2018; Padilla et al., 2017), these provide regional-to-global commission/omission error statistics which need to be interpolated with a statistical model of the measurement process to provide explicit spatiotemporally dense uncertainties (such as is done in GFED4). Specifying and then parameterising such models spatially and temporally is a considerable challenge. Instead, the presented triple collocation (TC) error model provides a data-driven method to independently and automatically estimate uncertainties in three global burned area products post hoc, and in a manner suitable for inclusion as part of stage four validation campaigns.

A feature of the TC analysis shown here is the large relative uncertainties across croplands and shrublands globally. The large relative uncertainties in shrubland burning have not been previously highlighted for global satellite burned area products. A potential mechanism for this is a detection threshold associated with the limited and discontinuous fuel bed in shrublands. The limited vegetation density in shrublands will limit the magnitude of the radiometric burn signal pre-to-post fire – limiting

the change signal the algorithms use to classify burning. Combing the limited vegetation signal with the general sparseness of vegetation ground cover in shrublands will lead to this 'patchiness' of the burn signal which when observed at 500m will fall around the detection thresholds of the mapping algorithms considered here (Roy and Landmann, 2005). The large relative uncertainty for MCD45 recorded in Australian (primarily xeric) shrublands is potentially a feature of the limited performance of the algorithm over surfaces with bright soils (de Klerk et al., 2012; Roy et al., 2005). This is an interesting feature that represents a promising area for future research. Cropland burning has been a persistent problem for coarse resolution burned area products. Particular features which obscure detection in croplands are the transient nature of the burn signal before ploughing, and the highly fragmented nature of burning on the land surface. Given these circumstances, the ability to detect cropland burn scars from MODIS resolution data has been previously questioned (Hall et al., 2016). Zhu et al. (2017) indicated omission errors for the MCD64 product greater than 60% for small cropland fires. Similarly, MCD45 has been reported to considerably under-report cropland burning globally (Roy et al., 2008). However, discrepancies between the products are likely to still be driving the TC uncertainties, for example,, observed commission errors by MCD64 for harvesting in Eurasia and MCD45 in Australia (Humber et al., 2018; Giglio et al., 2009). It remains an open question whether the higher spatial resolution available in the FireCCI50 products improves performance over croplands, with some evidence that it might (Chuvieco et al., 2016). The FireCCI50 product detects the greatest magnitude of cropland burning globally and has the smallest relative uncertainties of the three products. Future studies may be better able to indicate whether the increase in spatial resolution has produced this.

Previous, validation activities have indicated that satellite-derived burned area products typically perform best in regions where fire activity is more prevalent (Padilla et al., 2015). We also find that the smallest relative uncertainties are typically found in the frequently burning savannas and grasslands of Africa, Australia and South America. Nevertheless, relative uncertainties in burned area estimates for these regions were found to be in excess of 8-10%. Given the predominance of fire activity in these areas, they contribute considerably to the uncertainty on reported global burned area. In areas with more infrequent burning or more barriers to detection, relative uncertainties were found to be higher. In such circumstances, the particular limitations of each detection algorithm are most likely to drive the differences observed. For example, differing observational requirements of the products drives large uncertainties in equatorial Asia (EQAS) where persistent cloud reduces the mapped area of all algorithms. The MCD45 algorithm has been found to suffer uniquely in cloudier regions due to the greater sampling requirement of the algorithm as well as over-restrictive cloud masking conditions (Roy et al., 2002; Humber et al., 2018; Giglio et al., 2010). Changes made to the MCD64 Collection 6 product, including relaxations on cloud masking have increased the mapped area in these cloudier regions (Giglio et al., 2018).

Globally, MCD64 reports the greatest burned area $3.76 \pm 0.15 \times 10^6$ km$^2$; followed by FireCCI50 $3.70 \pm 0.17 \times 10^6$ km$^2$ and then MCD45 $3.31 \pm 0.18 \times 10^6$ km$^2$. In terms of the global agreement between products, figure 11 shows the distribution of mean annual burned area for the three products. A higher level of agreement between the FireCCI50 and MCD64 products can be observed with the two products agreeing well within one standard deviation. The MCD45 product disagrees most with the MCD64 product and slightly less with the FireCCI50 product. The three products overlap within two standard deviations. Even so, the degree of discrepancy on global burned area estimates would indicate that the previously used confidence bounds (i.e. from the range of products (Rabin et al., 2017)) provide an under-estimate in the global burned area uncertainty.

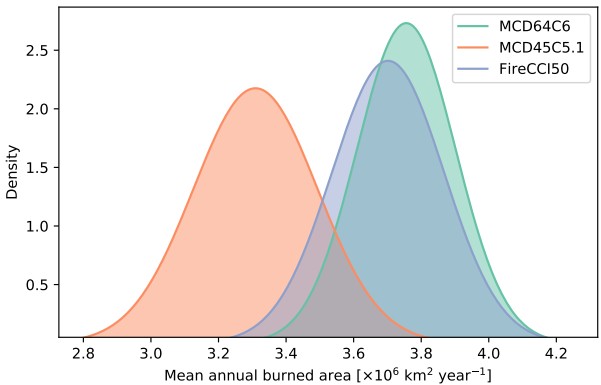

**Figure 11.** Constraints on global mean annual burned area $\mathrm{km}^2/\mathrm{year}$ provided by the three products.

Estimates of the mean annual burned area from the three products agree within their respective uncertainties in around 60% of valid TC-estimates. Nevertheless, while estimates are consistent, regional estimates remain poorly constrained by the products considered. Uncertainties in excess of 10% are found for all products in at least one land cover, including uncertainties >24% for MCD45 in shrublands, 11% for MCD64 in croplands and 13% in shrublands for FireCCI50. Regional uncertainties are often larger than these figures, with relative uncertainties in excess of 100% for MCD45 in croplands and grasslands in central America and boreal Asia; and for forests in Europe and boreal North America. Uncertainties larger than 100% for MCD64 are also found in forests and croplands in boreal and central Asia. FireCCI50 also has relative uncertainties >100% for croplands and forests in Australia, boreal North America and Europe. As these products are often also used at national to regional scales, it is important to consider the reliability of the current products at these scales (Liu et al., 2018; Zhu et al., 2017; Roy and Boschetti, 2009). The uncertainty estimates here are therefore useful for these users to discern any limitations of products at the appropriate scale. While the TC-estimated uncertainties can not directly provide information on uncertainties at the pixel level, we would also encourage users to consider the quality assurance (QA) information provided in these products.

The presented TC uncertainties have many uses. The uncertainties could, for example, be used to drive development and refinement of parameters in dynamic vegetation models (DVGMs) related to fire processes or improve optimisation routines for parameter selection (Poulter et al., 2015; Forkel et al., 2019). They could also be used to better constrain uncertainties on emission estimates derived from 'bottom-up' inventory approaches (Randerson et al., 2012; French et al., 2004; Knorr et al., 2012; Van Der Werf et al., 2017). Explicit uncertainties per observation additionally allow for the development of more advanced assimilation of the satellite observations into models through mathematical frameworks in data assimilation. Similarly, they open up the ability to calibrate model parameters $\mathbf{x}$ against observations of burned area. For example, assume a DGVM has a fire model that predicts burned area at a time $t$ ($\mathrm{BA}_{\mathrm{model}}(t)$) as a function of e.g. meteorological drivers, vegetation parameters and some fire-related parameters $I$ (e.g. Thonicke et al. (2010); Mangeon et al. (2016)):

$$H(\mathbf{x}, I, t) = \mathrm{BA}_{\mathrm{model}}(t). \tag{13}$$

Under the assumption that the burned area estimates are normal, one could derive the (log)likelihood function $L(\mathrm{BA}_{\mathrm{obs}} \mid \mathbf{x})$, which can be written as:

$$L(\mathrm{BA}_{\mathrm{obs}} \mid \mathbf{x}, t) \propto \frac{[H(\mathbf{x}, I, t) - \mathrm{BA}_{\mathrm{obs}}(t)]^2}{2\sigma_{\mathrm{TC}}(t)^2}. \tag{14}$$

Minimisation of this function would result in the parameters that provide a closes fit to the observations, weighted by how much one could trust these observations.

## 7  Conclusions

The wide application and interpretation of remote sensing products of burned area require explicit estimates of the uncertainties of these products. This paper has presented theoretical uncertainties for three global satellite-derived burned area products. A triple collocation (TC) error model was applied to produce unique, near-global, uncertainties for the MCD64 Collection 6, MCD45 Collection 5.1, and FireCCI50 burned area products. While products were found to provide consistent estimates in a majority of the sampled global fire extent, the constraint on burned area in many regions was found to be poor with uncertainties in each product exceeding 8-10% in the most burned regions. Uncertainties on burned area in regions with less burned area were also found to be considerable. Individual products were shown to have uncertainties exceeding 100% in specific regions and land covers. The present study would suggest that previous estimates of uncertainty in global burned area from satellite products appear to be under-estimates. Users of these products should therefore be aware of the uncertainties both in the limited constraint on burned area even from multiple products, and the regional and land cover specific differences in product confidence as provided by these uncertainties.

*Data availability.*  The TC estimated uncertainties are available at https://catalogue.ceda.ac.uk/uuid/2d9162f949e042adbdd6ec82c910ee5b.

*Author contributions.*  JB designed the study and performed the analysis with input from MD, JGD and PL. JB wrote the manuscript with contributions from all authors.

*Competing interests.*  The authors declare no conflict of interest.

*Acknowledgements.*  The authors were supported by the Natural Environment Research Council's (NERC) (Agreement PR140015 between NERC and the National Centre for Earth Observation, NCEO). JGD and PL would like to acknowledge financial support from the European Union Horizon 2020 research and innovation programme under grant agreement No 687320 MULTIPLY (MULTIscale SENTINEL land surface information retrieval Platform). We thank colleagues involved in ESA FireCCI Phase 2 for helpful discussions.

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
