# Peer review of "Theoretical uncertainties for global satellite-derived burned area estimates"

_Biogeosciences, 2019_

## Referee Comment (RC1) · Anonymous Referee #1 · 13 May 2019

The study provides uncertainty estimates for satellite burned area datasets. The methods are plausible and certainly go beyond any approach that has been described before. The manuscript is well written and requires only in few places some clarifications. Understanding uncertainties in datasets is crucial to apply them and to extract information that is valid. The manuscript does however provide only few background on how these uncertainty estimates can be used. The method also only represents random errors. This is a big limitation as the true burned area is likely far higher than what is estimated with these coarse resolution datasets. A recent study using Landsat data estimates an 80% higher burned area for Africa (Roteta et al. 2019). This indicates that the systematic errors are high and global burned area estimates of all globally available datasets are likely far too low. However, the relative differences of uncer-

tainties between regions and between land cover types may be very useful in spite of the lack of including systematic errors in the uncertainty estimates. Including the recent publication (Roteta et al. 2019) in the discussion and the consequences for the interpretationf of the uncertainties presented here is necessary. A broader discussion of how such uncertainties can be used in modelling studies and data analysis could strongly increase the impact of the paper.

I have a number of specific comments that hopefully improve the manuscript:

p.1, l. 1/2: essential for the scientific application of these datasets.. They are already used in science so please be more specific on why uncertainties are important.

p.1,l. 9: how about data analysis studies?

p.1,l. 5-6: how are these uncertainty measures to be interpreted given new dataproducts that indicate 80% higher burned area in Africa?

p.1,l. 12: looks like a unit (m-1 km) probably change to 250-1000m, or anything else more precise.

p. 4 l. 3: total burned area of what? the gridcell? The method also assumes that the error scales with the magnitude of the burned area, which is mentioned on p. 5 (heteroscedasticity). Here some restructuring would be useful.

p.4 l. 5 : Another arising concern is that the standard error maybe not only scales with the magnitude of burned area but other factors could be important. For instance land cover (e.g. woody cover that could hide subcanopy fires, cropland cover that usually is exposed to small sized fires, cloud cover, or other failures of the sensor or data transmission).

p.4 l. 7: how large are they, how to they differ from GFED

p.4 l. 21: Rabin et al. 2017: is this the correct ref? This is a model documentation paper.

p.4 l. 21-22: I don't understand what you want to say here?

p.4 l. 23: how are these uncertainties estimated?

p.5 l. 20: What is the distribution of the errors?

p.5 l. 25: the random errors or the standard deviation of the random errors is correlated with the magnitude?

p.5 l.26: Figure 1 could be changed to show the standard deviation over the products vs. the mean. That would more clearly show the heteroscedasticity and also the homoscedasticity for the log transformed data.

p.6,l. 12,p.7 l.1: move the "C" to diretly follow "sample covariance matrix"

p.7 l. 11-15: how about using the square root or maybe 10th root transformation to keep the 0 values?

p.7 l.1: Why are the annualised uncertainties of interest. please provide an overview on how uncertainties can be used and how the uncertainties are used by users at some place in the manuscript (maybe introduction).

p.8. l. 5: what about temporal auto-correlation of errors?

p.8 l. 12: total burned area of individual years or a multiyear mean?

p.8 l. 14: reason for using land cover type classification is that you assume that the local fire behaviour is driven by land cover type? Please clarify and add a reference for this assumption.

p.9 l. 2: change to "4) savannas"

p.9 l. 13-14: maybe add that no assumptions on the error structure are necessary in that way.

p.9 l.18-19: what does it actually mean if the random errors are larger than 100%? can the data be used for anything at all? Or is there no information content in these parts

then?

p.9 l. 33: As far as I know the FireCCI50 dataset has only been released last year, are you sure it is included in Humber et al. 2018? In their description it says the product is based on MERIS.

p.10 l. 4: what exactly is consistent?

p.10 l.6-7: maybe a root transformation could be advantageous then.

p.11 l. 9: mean annual burned area?

p.12 l. 1: why are the uncertainties in shrublands high? has this been documented before? the higher uncertainty in croplands is well known due to the smaller fire size. But what could be a reason for high uncertainty in shrublands?

p.12 l.8: 8-10% seems low, given that the contribution of small fires, which are suggested to be mostly cropland fires is around 100 Mha (Randerson et al. 2012). And what does this estimation of the random error mean for the global extent of cropland burning? Systematic errors are not considered and the main effect of the small sized fires should be a systematic underestimation of the burned area on croplands.

p.12 l. 13-15: First sentence says lower uncertainties in BOAS, second sentence says larger uncertainties in BOAS. Please clarify.

p.13, l. 3: what is the shared uncertainty envelope and where can it be seen?

p.13 l. 4: now the relative uncertainties for savannas are larger than for croplands?

p.13 l. 7: "region" is double.

p.13 l.11-13: I don't understand: GFED4 is exclusively derived from MCD64 and tehn next sentence says GFED4ustilised the older MCD64 col. 5.1. Please clarify.

p.17 l. 3: " as evidenced..." I do not understand, can you explain this better?

p.18 l.12-13: do you mean errors of your error estimates or the estimated errors?

p.19 l. 11: but the uncertainties for shrublands were largest?

p. 20 l. 1: globally there should be still a large underestimation due to the coarse resolution. for instance Roteta et al. (2019) recently estimated a 80% higher burned area in Africa. How does this influence the interpretation of the here presented uncertainties. The true global burned area is then very likely outside of the range of global burned areas presented here, as only random errors are captured.

p.20 l. 7: I can't find confidence bounds presented in Rabin et al. 2017.

p.21: I think the conclusions as well as the discussion chapter should provide information how and for what the uncertainties can be used. You write theoretical uncertainties, but they are meant to be used in practice right?

p.21 l. 11: what do you mean with unique error characteristics? the regional and land cover specific differences in uncertainties?

References:

Randerson, J. T., Chen, Y., van der Werf, G. R., Rogers, B. M. and Morton, D. C.: Global burned area and biomass burning emissions from small fires, J. Geophys. Res. Biogeosciences, 117, G04012, doi:10.1029/2012JG002128, 2012.

Roteta, E., Bastarrika, A., Padilla, M., Storm, T. and Chuvieco, E.: Development of a Sentinel-2 burned area algorithm: Generation of a small fire database for sub-Saharan Africa, Remote Sens. Environ., 222, 1–17, doi:10.1016/j.rse.2018.12.011, 2019.

---

## Referee Comment (RC2) · Anonymous Referee #2 · 31 May 2019

General Comments:

Brennen et al. present an estimate of the uncertainties of global burned area estimates for three products for the period spanning January 2001 to December 2013. The uncertainty estimates are based on the triple collocation (TC) analysis model which has been used in other fields including wind speed and soil moisture estimation. Results from this study could be useful to the modelling community. The structure of the paper suits the research well, and the manuscript nicely summarizes the state of burned area products and the methods used for creating the products in question (MCD64A1 C6, MCD45A1 C5.1, FireCCI50). However, the authors have missed some of the recent advances in product validation and generation which might make this manuscript out of date already.

[Figure]

Beginning with the product selection, two of the products (MCD45A1 C5.1 and FireCCI50) have been replaced at this point – the former by the MODIS Collection 6 MCD64A1 product and the latter by FireCCI51 (https://geogra.uah.es/fire_cci/). As such, the use of a deprecated product such as MCD45A1 C5.1 seems odd assuming that the Collection 6 implementation should be an improvement over the outgoing product. Recognizing that the triple collocation method requires a third dataset, a current operational product such as the Copernicus Burnt Area (https://land.copernicus.eu/global/products/ba) could have been implemented.

Broadly, there should be a discussion of the influence of data set selection on the results of the uncertainty indicators. Several factors are of concern in this regard: – The accuracy of burned area products are generally "low," where omission errors ranging from 60% to 80% and commission errors range from 30% to 60% for three global products which included FireCCI50 and MCD64A1 C6 (according to Chuvieco et al., 2018). How does the accuracy influence the result of the TC analysis, given that the accuracy of the products is unknown for the purposes of this study? Should the reader interpret the results as being specific to these 3 products? – Considering the high rate of omission errors, is it not likely that the requirement that all three products identify burning in a cell is overly restrictive? What if one is wrong and two are not? It would be helpful to know how many of the 40% of cells (Page 7 Line 13) had burned area identified by at least one product. – Related to the previous point, a value of 0 burned area can be a correct classification. Is it valid to throw out the value 0 simply because of the log transformation? It is not a "no data" value. – Roteta et al. (2019) claim that burned area estimates in Africa are more than 80% higher when using 20m Sentinel-2 data compared to MCD64A1. Their result is incompatible with the results of this work, so there needs to be a more nuanced explanation of what uncertainty is in the context of this study given that comparison of two products at 250m resolution is not the same as one product at 250m and another at 20m.

A more important issue is the recent advances in burned area product validation which

were not accounted for in this manuscript. For example, in P2 L9-11, the (uncited) claim is made that "Even the largest and most sophisticated validation datasets correspond to only a small sampling of global fire activity, and it is not clear whether this is sufficient information to build an understanding of uncertainties at global and decadal scales". The work done by Boschetti et al., 2016; Padilla et al., 2015; and Padilla et al., 2017 show that uncertainty can, in fact, be identified at the global scale using stratified random sampling. On P1 L17, it is stated (again uncited) that the "true information content of such datasets is still unquantified," yet the uncertainties of the FireCCI50 and MCD64A1 products are provided as part of the accuracy assessment done by Chuvieco et al. (2018) as well as within the FireCCI50 gridded product itself (P4 L15). The references to other works which call for the availability of uncertainty data (e.g. Mouillot et al., 2014; Rabin et al., 2017; Yue et al., 2014; Knorr et al., 2014) pre-date the work which has been done through more current validation exercises such as that in Chuvieco et al. (2018) and Roteta et al. (2019).

Finally, the manuscript needs more context to explain why this work is necessary, especially given that there are Stage 3 burned area validation datasets which can provide estimates of uncertainty in the burned area measurements. The CEOS LPV guidelines for validation stages are referenced in the Discussion, and while the validation stage definitions are somewhat vague for stage 3 (https://lpvs.gsfc.nasa.gov/), "uncertainty" has typically been understood to refer to accuracy with an associated uncertainty accompanying the accuracy estimate. While the TC method in this manuscript can provide uncertainty estimates, perhaps even compatible with Stage 4, the uncertainty is presented independent of the accuracy of the data set – a user cannot use the uncertainty information alone to know how likely a given pixel is to be correctly labeled. This is of less concern to the modelling community, but should be addressed nonetheless.

Specific Comments:

– The abstract needs to be rewritten. The first sentence ends with a dangling participle ("these datasets" refers to nothing); the study period should be included in the abstract; the sentence about the uncertainty estimates is unclear – at minimum it should note that the estimates are per year, but the phrase "Theoretical uncertainties indicate constraints..." is unnecessarily complicated given that the values are simply burned area estimates with uncertainty; "product" should be "products"; why are Africa and Australia singled out in the abstract?

– A definition of uncertainty should be provided to distinguish between uncertainty in total burned area vs (for example) temporal uncertainty in day of burning. This is also important in light of the findings of Roteta et al. (2019) whose results are incompatible with these using the conventional understanding of uncertainty.

– P2 L7 "validation exercises": Some of the previously referenced studies are intercomparisons, not validation exercises - the former does not imply accuracy assessment. For example, in Humber et al. (2018), no assumption is made that any one product is correct and in fact it is possible that all four products are incorrect for any given burn.

– P3 L22-23 "Simon et al. [...]": This does not need to be included, all algorithms have parameters which lead to commission/omission errors.

– P7: How is the aggregation affected by temporal uncertainty, such as that indicated by the MCD64A1 SDS or the nominal 8-day uncertainty of the MCD45A1 product? In theory even a 1-day shift in burn date detection could lead a cell to be excluded erroneously from a 16-day period and the temporal uncertainty of MCD45A1 is in fact greater than half of the compositing period. Generally, given that this is a paper about uncertainty, it would be good to incorporate the temporal uncertainty in the measurements somehow.

– P8: Generally, it would benefit the reader to have a discussion of the fire seasonality – the calendar year has been shown to be a fairly bad cutoff period for burned area. In Figure 3, the reader would benefit from knowing that the peaks in uncertainty correspond to the peak of the burning season in Australia. The legend for the figure should also indicate that this figure refers to Australia, and the figure on the right is missing

the x-axis labels.

– P12 L8: The problem with the definition of uncertainty is very evident here ("[. . .] cropland burning with relative uncertainties of 8-10%.") The products generally agree with each other about cropland burning, however they are all severely underestimating the total amount (See Hall et al., 2016 which demonstrated MCD45A1 and MCD64A1 underestimate agricultural burning by > 90%). This illustrates that the uncertainty presented here is relative to the other products, underscoring the need for ground data as a baseline for comparison.

– P13: The comparison the MCD64A1 C5.1 might not be relevant, many things about the product were changed and there is not a flat 26% increase in burned area globally – some regions increased significantly more than others, and the detection rate of small fires is significantly higher in the Collection 6 product. Perhaps a better test of the TC method would be to replace the Collection 6 product with Collection 5.1 for the purpose of comparing the result.

– P19 L5-7 "the majority of current burned area products have only achieved stage two validation": What are the current burned area products referred to in this sentence? Of publicly available operational (global) products, three come to mind (FireCCI51, MCD64A1, Copernicus Burnt Area), and of those two were validated at Stage 3 in Chuvieco et al. (2018). The reference to Padilla et al. (2014) is actually a strategy for Stage 3 validation (not Stage 2), and the reference should be expanded to include Boschetti et al. (2016) and Padilla et al. (2017), both of which improve upon the temporal robustness of the sample necessary to provide accuracy and uncertainty estimates through time.

– P20 L17-18: It seems the results are relevant to modelers at coarse resolutions. How would a user implement this work at finer scales?

– P21: How should a user implement the information from this work? What about reconciling the differences with works like Roteta, who indicated burned area totals in

Africa are well outside of the bounds of uncertainty presented in this work?

Technical Comments:

– P5 L30: Should refer to eq. 2-4. – P6 Figure 1: Typo in legend of figure on the left.
– P7 L1: Specify 1 degree at the equator. – P11 L5: "product's" should be "products"
– P13 L32-33: The last sentence is a fragment. – P19 L4: The reference to Giglio et
al., 2006b should be with "GFED4" – P19 L17: Remove parenthesis around Zhu et al.
– P20 L16-17: References should be in parenthesis.

---

## Author Comment (AC1) · 8 Jul 2019

**AC:** We thank the reviewer for their useful comments on the MS. Below we address their concerns and provide revisions to the MS.

**Reply to general Comments**

**RC1:** The study provides uncertainty estimates for satellite burned area datasets. The methods are plausible and certainly go beyond any approach that has been described before. The manuscript is well written and requires only in few places some clarifications. Understanding uncertainties in datasets is crucial to apply them and to extract information that is valid. The manuscript does however provide only few background on how these uncertainty estimates can be used.

**AC:** We thank the reviewer for these kind comments. We agree that the discussion on how these uncertainty estimates can be used is too limited. To address this we have added a paragraph to the discussion proposing ways in which the uncertainties can be used by users: "While the TC-estimated uncertainties can not directly provide information on uncertainties at the pixel level, we would also encourage users to consider the quality assurance (QA) information provided in these products. The presented TC uncertainties have many uses. The uncertainties could, for example, be used to drive development and refinement 35 of parameters in dynamic vegetation models related to fire processes or improve optimisation routines for parameter selection (Poulter et al., 2015; Forkel et al., 2019). They could also be used to better constrain uncertainties on emission estimates derived from 'bottom-up' inventory approaches (Randerson et al., 2012; French et al., 2004; Knorr et al., 2012; Van Der Werf et al., 2017). Explicit uncertainties additionally allow for the development of more advanced assimilation of the satellite observations into models through mathematical frameworks in data assimilation."

**RC1:** The method also only represents random errors. This is a big limitation as the true burned area is likely far higher than what is estimated with these coarse resolution datasets. A recent study using Landsat data estimates an 80% higher burned area for Africa (Roteta et al. 2019). This indicates that the systematic errors are high and global burned area estimates of all globally available datasets are likely far too low. However, the relative differences of uncertainties between regions and between land cover types may be very useful in spite of the lack of including systematic errors in the uncertainty estimates. Including the recent publication (Roteta et al. 2019) in the discussion and the consequences for the interpretation of the uncertainties presented here is necessary. A broader discussion of how such uncertainties can be used in modelling studies and data analysis could strongly increase the impact of the paper.

**AC:** We agree that the ability of the TC method to only account for random errors is a limitation of the method. Systematic errors originating primarily from missing small fires in the coarse resolution products will ultimately inflate the total uncertainty in the

products. We would therefore regard the estimated uncertainties as providing a lower bound on the total uncertainty, in the absence of systematic errors (with the view that Total uncertainty=systematic + random errors). Given this, we agree with the view that the relative differences between regions and land cover types may actually be more useful for some users and still represents the most granular estimate of uncertainties available for these products.

We have included an additional section about this into the considerations of the TC method (section 5). "We also stress that the uncertainties estimated with the TC method likely represent a lower bound on the true uncertainties of these products. The TC measurement model can only explicit estimate random errors but not the likely systematic errors (i.e. bias) present in the data products. The under-estimation bias observed for these coarse-resolution products in validation studies indicates that the products likely have considerable systematic errors. Chuvieco et al. (2018) have estimated that the FireCCI50 product has global omission errors of 70% and MCD64C6 62%, which are partially balanced by commission errors of 50% and 35% respectively. Roteta et al. (2019) also indicated that a higher spatial resolution 20m burned area product provided 80% more burned area than the MCD64C6 product for sub-Saharan Africa, indicating considerable biases in coarse-resolution products. Users should be aware therefore that the likely systematic biases in coarse resolution products mean that the TC uncertainties provide a lower bound on the true uncertainty."

**Reply to specific comments**

**RC1:** p.1, l. 1/2: essential for the scientific application of these datasets.. They are already used in science so please be more specific on why uncertainties are important.
**AC:** We have clarified this in the abstract to reinforce that the uncertainties are "essential for evaluating the quality of these products and comparison against modelled estimates of burned area".

**RC1:** p.1,l. 9: how about data analysis studies?
**AC:** We have added reference to data analysis studies.

**RC1:** p.1,l. 5-6: how are these uncertainty measures to be interpreted given new data products that indicate 80% higher burned area in Africa?
**AC:** We think this is addressed by the discussion about systematic errors above.

**RC1:** p.1,l. 12: looks like a unit (m-1 km) probably change to 250-1000m, or anything else more precise.
**AC:** This is changed to (250m-1000m).

**RC1:** p. 4 l. 3: total burned area of what? the gridcell? The method also assumes that the error scales with the magnitude of the burned area, which is mentioned on p. 5 (heteroscedasticity). Here some restructuring would be useful.
**AC:** We have clarified this as: "the aggregated burned area in the grid cell".

**RC1:** p.4 l. 5 : Another arising concern is that the standard error maybe not only scales with the magnitude of burned area but other factors could be important. For instance land cover (e.g. woody cover that could hide subcanopy fires, cropland cover that usually is exposed to small sized fires, cloud cover, or other failures of the sensor or data transmission).
**AC:** We think this is a good point and a potential limitation of that method. We've addressed this by adding an additional paragraph: "An additional limitation of the regional enumeration of $c_B$ is that it must replicate contributions from additional uncertainty sources. These will be features such as variations in cloud cover obscuring burned area detection, and uncertainties arising from variations in the distribution and local mixture of vegetation type. This variability will alter the value of $c_B$ within each region."

**RC1:** p.4 l. 7: how large are they, how do they differ from GFED
**AC:** this has been clarified with reference to the 103 validation tiles used in that paper.

**RC1:** p.4 l. 21: Rabin et al. 2017: is this the correct ref? This is a model documentation paper
**AC:** yes, Rabin et al. 2017 refer to: "There are multiple datasets available for some of these properties, including, for example, burned area. Padilla et al. (2015) have

shown that currently available burned area products differ considerably both in terms of global total and at a regional scale. Differences between datasets effectively define the current range of uncertainty in observations, and this level of uncertainty needs to be taken into account when evaluating model performance." Page (1190)

**RC1:** p.4 l. 21-22: I don't understand what you want to say here?
**AC:** Thanks, we have rephrased this section to (hopefully) improve clarity.

**RC1:** p.4 l. 23: how are these uncertainties estimated?
**AC:** Le Page et al. (2015) detail that these are provided based on considering the papers for GFED/MCD45 and also comparing versions of GFED (pg. 895). We added "based on an inspection of the GFED data" to the manuscript.

**RC1:** p.5 l. 20: What is the distribution of the errors?
**AC:** these are considered here to be normally distributed. We have added "are considered to be normally distributed".

**RC1:** p.5 l. 25: the random errors or the standard deviation of the random errors is correlated with the magnitude?
**AC:** The standard deviation of the random errors. The random error model is formulated as normal distribution such that the errors are drawn from $N(0, \sigma)$. The multiplicative model deals with the characteristic that $\sigma = f(BA)$. We have clarified this in the manuscript.

**RC1:** p.5 l.26: Figure 1 could be changed to show the standard deviation over the products vs. the mean. That would more clearly show the heteroscedasticity and also the homoscedasticity for the log transformed data.
**AC:** Thanks, this is a good suggestion for figure 1. We have changed figure 1 to now plot mean over the products (x) vs individual product (y) and also the standard deviation of the products scaling with x. This makes the heteroscedasticity/homoscedasticity of the transform more apparent.

**RC1:** p.6,l. 12,p.7 l.1: move the "C" to directly follow "sample covariance matrix"
**AC:** Thanks, done.

**RC1:** p.7 l. 11-15: how about using the square root or maybe 10th root transformation to keep the 0 values?
**AC:** We thank the reviewer for this suggestion. Various transforms were also considered but an unfortunate feature of transforms other than the log transformation is the complication of the triple collocation model. The multiplicative model as phrased works because the log-transformation provides a multiplicative error which is linear in log-space. Further transforming a square-root transformed linear triple collocation such as $\sqrt{x} = \alpha + \beta\sqrt{T} + \epsilon$ back into real units (km$^2$) does not equate to a model in which the error $\epsilon$ fulfils the properties of being multiplicative, or indeed a random error component.

**RC1:** p.7 l.1: Why are the annualised uncertainties of interest? please provide an overview on how uncertainties can be used and how the uncertainties are used by users at some place in the manuscript (maybe introduction).
**AC:** We found that annualised estimates provided an efficient method to summarise regional disparities most clearly in a visual manner (e.g. figure 7). The actual uncertainties are provided for each 16-day period in the observational record (2001-2013) of the products, which is being registered with an online data repository. Annual burned area is also generally the focus of previous inter-comparison studies such as Humber et al. (2018) and also the papers describing the products e.g. Giglio et al. (2018). We agree that more information should be provided on how these uncertainties could be used and have added a section on this to the discussion detailed earlier. We have also extended the brief section on user requirements for uncertainties in the introduction (Pg2, L11).

**RC1:** p.8. l. 5: what about temporal auto-correlation of errors?
**AC:** we agree with the reviewer that an understanding of the auto-correlation of the uncertainties would be useful. However it is not easy to estimate this auto-correlation without a full treatment of the uncertainties in burned area at the pixel scale (i.e. including the temporal uncertainty which is only available for MCD64) and how this could be properly aggregated to the grid-scale burned area. Unfortunately the triple colloca- tion method as formulated is not able to formulate auto-correlation of errors but also assumes no correlation in errors between products.

**RC1:** p.8 l. 12: total burned area of individual years or a multiyear mean?
**AC:** thanks we have now clarified this by adding "for each individual year".

**RC1:** p.8 l. 14: reason for using land cover type classification is that you assume that the local fire behaviour is driven by land cover type? Please clarify and add a reference for this assumption.
**AC:** This is a good point and variations associated more with fire characteristics (or fire pyromes) may be better. We chose to focus on the combination of the GFED regions and broad land cover classes because this formulation has been used previously for several papers and would hopefully be familiar to readers. Some examples are Giglio et al. 2010, 2013 for GFED which uses the regions and these land cover super classes. The papers describing MCD64 also use this formulation (Giglio et al. 2018) and the paper for FireCCI MERIS (Alonso-Canas et al. 2015).

**RC1:** p.9 l. 2: change to "4) savannas"
**AC:** changed.

**RC1:** p.9 l. 13-14: maybe add that no assumptions on the error structure are necessary in that way.
**AC:** thanks. We have added "while requiring no additional assumptions about the error structure".

**RC1:** p.9 l.18-19: what does it actually mean if the random errors are larger than 100%? can the data be used for anything at all? Or is there no information content in these parts then?
**AC:** This would indicate yes that in these locations the precision of the burned area is actually less than the uncertainty. This most obviously arises when the three products

provide very divergent estimates such that the products show little agreement on the magnitude of burning. In such cases the products should be trusted least. To provide more information on this we have added: "This would indicate that the level of agreement between the products is lower than the precision of the products".

**RC1:** p.9 l. 33: As far as I know the FireCCI50 dataset has only been released last year, are you sure it is included in Humber et al. 2018? In their description it says the product is based on MERIS.
**AC:** Yes this is a mistake – Humber et al. 2018 analyse FireCCI MERIS. We corrected this by referring only to MCD64 in reference to Humber et al. 2018.

**RC1:** p.10 l. 4: what exactly is consistent?
**AC:** consistency here means that the distributions of burned area for each product show overlap – i.e. the products agree within their uncertainties. We have clarified this as "consistent within the uncertainties".

**RC1:** p.10 l.6-7: maybe a root transformation could be advantageous then.
**AC:** See the comment above about the problems of a root transformation for the triple collocation error model.

**RC1:** p.11 l. 9: mean annual burned area?
**AC:** Thanks, corrected.

**RC1:** p.12 l. 1: why are the uncertainties in shrublands high? has this been documented before? the higher uncertainty in croplands is well known due to the smaller fire size. But what could be a reason for high uncertainty in shrublands?
**AC:** We also found this an interesting finding that (as far as we are aware) has not been documented before. Our primary view is the likely difficulty of detection here from 500m data arising from burn 'patchiness' as a response of the limited and discontinuous fuel bed in shrublands. The much lower vegetation density in shrublands will limit the magnitude of the radiometric burn signal pre-to-post fire – limiting the change signal the algorithms use to classify burning. Combing the limited vegetation signal

with the general sparseness of vegetation ground cover in shrublands will lead to this 'patchiness' of the burn signal which when observed at 500m will likely fall around the detection thresholds of these burned area mapping algorithms (for example see Roy Landmann, 2005). The aggregated uncertainties for shrublands also hides the fact that the uncertainties for 'hot' (xeric) and 'cold' (tundra etc.) shrublands varies quite considerably. The large relative uncertainty for MCD45 recorded in Australia (primarily xeric) shrublands is potentially a feature of the limited performance of the algorithm over surfaces with bright soils (Roy et al., 2005; de Klerk et al., 2012). This is not replicated for 'cold' shrublands the same manner which generally have darker soils. We have added this to the discussion of the paper:

"The large relative uncertainties in shrubland burning have not been previously highlighted for global satellite burned area products. A potential mechanism for this is a detection threshold associated with the limited and discontinuous fuel bed in shrublands. The limited vegetation density in shrublands will limit the magnitude of the radiometric burn signal pre-to-post fire – limiting the change signal the algorithms use to classify burning. Combing the limited vegetation signal with the general sparseness of vegetation 30 ground cover in shrublands will lead to this 'patchiness' of the burn signal which when observed at 500m will fall around the detection thresholds of the mapping algorithms considered here (Roy and Landmann, 2005). The large relative uncertainty for MCD45 recorded in Australian (primarily xeric) shrublands is potentially a feature of the limited performance of the algorithm over surfaces with bright soils (de Klerk et al., 2012; Roy et al., 2005). This is an interesting that represents a promising area for future research."

**RC1:** p.12 l.8: 8-10% seems low, given that the contribution of small fires, which are suggested to be mostly cropland fires is around 100 Mha (Randerson et al. 2012). And what does this estimation of the random error mean for the global extent of cropland burning? Systematic errors are not considered and the main effect of the small sized fires should be a systematic underestimation of the burned area on croplands.

**AC:** We agree that croplands will have higher systematic errors due to omission errors

for some products. We would argue that it is difficult to be sure about the likely direction of this effect however due to observed commission errors by MCD64 for harvesting in Eurasia and MCD45 in Australia (Humber et al. 2018). Because of these discrepancies in the response of products we could realistically expect that at least some of the systematic error is present in the random errors of the products. To comment on this we have added to the text: "However, discrepancies between the products are likely to still be driving the TC uncertainties, for example" observed commission errors by MCD64 for harvesting in Eurasia and MCD45 in Australia (Humber et al., 2018; Giglio et al., 2009)"

**RC1:** p.12 l. 13-15: First sentence says lower uncertainties in BOAS, second sentence says larger uncertainties in BOAS. Please clarify.
**AC:** Thanks, this is clarified as: "Uncertainties for MCD45 are around two times larger in BONABOAS forests, and 40% larger for FireCCI50 in BOAS as compared to BONA forests. Alternatively, MCD64 has lower relative uncertainties in BONA compared to BOAS, with uncertainties 70% larger in boreal Eurasia."

**RC1:** p.13, l. 3: what is the shared uncertainty envelope and where can it be seen?
**AC:** This refers to the central BA estimates being within the standard errors of each product. Such that the distributions of each product overlap within 1 standard deviation. We have clarified this in text by substituting the "uncertainty envelope" for "for each product agreeing within the uncertainties estimated for all products".

**RC1:** p.13 l. 4: now the relative uncertainties for savannas are larger than for croplands?
**AC:** For northern Hemisphere (NHAF) and southern hemisphere Africa (SHAF) relative uncertainties in savannas are larger than croplands. To make this clearer we have rephrased this to: "The uncertainties are still considerable, however, with relative uncertainties for all three products largest in savannas and grasslands. In these land covers, relative uncertainties exceed 13% in NHAF and 8% in SHAF."

**RC1:** p.13 l. 7: "region" is double.
**AC:** Thanks.

**RC1:** p.17 l. 3: " as evidenced..." I do not understand, can you explain this better?
**AC:** this refers to the discontinuous patterns that can be seen in the probability field for FireCCI50. These most likely occur due to the compositing method used in the algorithm which determines the number of available observations for the retrieval of burned area. The same tesselation pattern can be seen in the ATDB for the algorithm on page 29. (https://www.esa-fire-cci.org/sites/default/files/Fire_cci_D2.1.3_ATBD-MODIS_v1.1.pdf We have clarified this in the manuscript by: "with the apparent pattern in unburned confidence values arising from the interpretation of the composited observations used within the algorithm."

**RC1:** p.18 l.12-13: do you mean errors of your error estimates or the estimated errors?
**AC:** This section refers to potential sources of correlations (ECCs) in the actual errors between products and the true burned area. Depending on the strength of these EECs, the assumptions of the triple collocation method may not be met. So this section explores whether the uncertainty estimates are likely to be "tainted" by ECCs. We've clarified that ECCs alter the quality of TC uncertainties in this paragraph.

**RC1:** p.19 l. 11: but the uncertainties for shrublands were largest?
**AC:** This is correct – shrublands did have larger relative uncertainties globally for all three products than croplands. To clarify this we have rephrased the first sentence to: "A feature of the TC analysis shown here is the large relative uncertainties across croplands and shrublands globally". We have then also added a discussion about the potential mechanisms for large shrubland uncertainties as detailed above.

**RC1:** p. 20 l. 1: globally there should be still a large underestimation due to the coarse resolution. for instance Roteta et al. (2019) recently estimated a 80% higher burned area in Africa. How does this influence the interpretation of the here presented uncertainties. The true global burned area is then very likely outside of the range of global

burned areas presented here, as only random errors are captured.

**AC:** We partially agree with the reviewer here. The underestimation by coarse resolution products detailed in Roteta et al. (2019) will ultimately mean that the true uncertainty on the coarse resolution products will be larger. This systematic error which we referred to earlier may exceed the random error for some regions. Because of this users should be aware that these uncertainty estimates represent a lower bound on the true uncertainty. We would also caution that while for some regions the systematic error > random error this may not be the case for all regions It has not been established how large the global underestimation will be with the additional consideration that a portion of every 500m pixel labelled burned in the products will only be fractionally burned. To address this point we added the section detailed above to Section 5 on systematic and random errors.

**RC1:** p.20 l. 7: I can't find confidence bounds presented in Rabin et al. 2017.

**AC:** Rabin et al. (2017, pg. 1190) refer to the "Differences between datasets effectively define the current range of uncertainty in observations, and this level of uncertainty needs to be taken into account when evaluating model performance."

**RC1:** p.21: I think the conclusions as well as the discussion chapter should provide information how and for what the uncertainties can be used. You write theoretical uncertainties, but they are meant to be used in practice right?

**AC:** We agree with the reviewer here. To address this we have added the paragraph detailed earlier to the discussion.

**RC1:** p.21 l. 11: what do you mean with unique error characteristics? the regional and land cover specific differences in uncertainties?

**AC:** Thanks, exactly that. To improve this we have added "and the regional and land cover specific differences in product confidence as provided by these uncertainties."
* * *

---

## Author Comment (AC2) · 9 Jul 2019

**AC:** We thank the reviewer for their considered comments. Below we detail our responses to their concerns including revisions we have made to the manuscript to hopefully address these.

**RC2:** Brennen et al. present an estimate of the uncertainties of global burned area estimates for three products for the period spanning January 2001 to December 2013. The uncertainty estimates are based on the triple collocation (TC) analysis model which has been used in other fields including wind speed and soil moisture estimation. Results from this study could be useful to the modelling community. The structure of the paper suits the research well, and the manuscript nicely summarizes the state of burned area

products and the methods used for creating the products in question (MCD64A1 C6, MCD45A1 C5.1, FireCCI50). However, the authors have missed some of the recent advances in product validation and generation which might make this manuscript out of date already.

**AC:** We thank the reviewer for their considered comments. We discuss these recent advances below in more detail.

**RC2:** Beginning with the product selection, two of the products (MCD45A1 C5.1 and FireCCI50) have been replaced at this point – the former by the MODIS Collection 6 MCD64A1 product and the latter by FireCCI51 (https://geogra.uah.es/fire_cci/). As such, the use of a deprecated product such as MCD45A1 C5.1 seems odd assuming that the Collection 6 implementation should be an improvement over the outgoing product. Recognizing that the triple collocation method requires a third dataset, a current operational product such as the Copernicus Burnt Area (https://land.copernicus.eu/global/products/ba) could have been implemented.

**AC:** We note that we did consider using the Copernicus Burnt Area product. The issue with this was that the Copernicus product covering the main period of the study has been decommissioned due to an artificial "decline in the amount of burned surface detected on a year by year basis". Please see: https://land.copernicus.eu/global/content/burnt-area-1km-spotvgt-unavailable. The newer Copernicus product derived from PROBA-V is only available from 2014. This would only provide at best 3 years of data (FireCCI50/51 ends in 2017). Given this limitation, the use of MCD45 was considered to provide a better long term record. At the time of writing the manuscript, the newer FireCCI51 was not available. Strengths and limitations of the MCD64 product have been highlighted in relation to the older MCD45C5.1 product. Most obviously differences in burned area detected in different regions: e.g. more burned area detected for MCD45C5.1 than MCD64C6 in Europe and the United States (Humber et al. 2018). We have added a discussion of this to the method section:

"The MCD45C5.1 product has now been deprecated by the Collection 6 MCD64 algorithm. The operational 1km Copernicus burned area product was also considered

however issues have been found in the product which has resulted in the product being withdrawn for re-processing (CopernicusWWW, 2019). The newer 300m Copernicus burned area product covers a more limited temporal span from 2014–present. In terms of data set selection the three chosen products represent the longest available combined satellite record."

**RC2:** Broadly, there should be a discussion of the influence of data set selection on the results of the uncertainty indicators. Several factors are of concern in this regard: – The accuracy of burned area products are generally "low," where omission errors ranging from 60% to 80% and commission errors range from 30% to 60% for three global products which included FireCCI50 and MCD64A1 C6 (according to Chuvieco et al., 2018). How does the accuracy influence the result of the TC analysis, given that the accuracy of the products is unknown for the purposes of this study? Should the reader interpret the results as being specific to these 3 products?

**AC:** In terms of data set selection, the three chosen products represent the longest available combined satellite record and so there are no other products that could be selected over that time frame. Consequently, yes users should interpret the results as being specific to the 3 products.

The comment about omission errors leading to a low accuracy refers to systematic errors (e.g. biases) in the products. A limitation of the multiplicative TC method is that it is only able to estimate random errors. The TC method therefore provides information on the precision of the grid cell burned area observations but not their accuracy/bias. With the view that the total uncertainty = systematic + random errors, the implication is that the TC estimated uncertainties provide a lower bound on the true uncertainties. To make this point clearer in the manuscript we have added the following discussion to the considerations of the TC method (Section 5):

"We also stress that the uncertainties estimated with the TC method likely represent a lower bound on the true uncertainties of these products. The TC measurement model can only explicit estimate random errors but not systematic errors (i.e. bias) present in the data products from fires which are e.g. undetectable due to the limitations in the observations. The under-estimation bias observed for these coarse-resolution products in validation studies indicates that the products likely have considerable systematic errors. Chuvieco et al. (2018) have estimated that the FireCCI50 product has global omission errors of 70% and MCD64C6 62%, which are partially balanced by commission errors of 50% and 35% respectively. Roteta et al. (2019) also indicated that a higher spatial resolution 20m burned area product provided 80% more burned area than the MCD64C6 product for sub-Saharan Africa, which while not providing a true validation indicates a considerable underestimation bias in coarse-resolution products. Users should be aware therefore that the likely systematic biases in coarse resolution products mean that the TC uncertainties provide a lower bound on the true uncertainty."

**RC2:** – Considering the high rate of omission errors, is it not likely that the requirement that all three products identify burning in a cell is overly restrictive? What if one is wrong and two are not? It would be helpful to know how many of the 40% of cells (Page 7 Line 13) had burned area identified by at least one product.

**AC:** We agree with the reviewer that this is a limitation of the multiplicative triple collocation method. The requirement stems from the necessary phrasing of the multiplicative model to achieve a linear normally-distributed additive error model in log-space. Other transforms such as the square root transform were considered due to their ability to contain 0 values. However, these transformations are generally not suitable multiplicative models when transformed back to the real line (e.g. burned area $km^2$). For example, transforming a square-root transformed linear triple collocation such as $\sqrt{x} = \alpha + \beta\sqrt{T} + \epsilon$ back into real units ($km^2$) does not equate to a model in which the error $\epsilon$ fulfils the properties of being multiplicative, or indeed a random error component.

We have clarified the 40% figure: Around 50% of cells had burned area identified by at least one product, though this figure is predominantly determined by cells with very little detected burned area by any product. For example, enforcing that any product has to have detected at least 10km$^2$ of burned area over the 13 years (or 40 MODIS pixels) reduces this figure to 30%. The method is able to sample the majority of the reported

fire activity by the products. Total burned area for 2001-2013 which do not have associated uncertainties is less than 0.5% of the total burned area of each product. We have clarified the 40% figure in the manuscript to indicate the sampling of global burned area by the TC method with: "The TC method is able to sample the majority of the reported fire activity by the products. Total burned area over the study period for cells which do not have associated uncertainties is less than 0.5% of the total burned area of each product. "

**RC2:** – Related to the previous point, a value of 0 burned area can be a correct classification. Is it valid to throw out the value 0 simply because of the log transformation? It is not a "no data" value.
**AC:** We agree with the reviewer but given that the overall effect is small (see discussion above) we feel that the additions made to the manuscript from the comment make this clear.

**RC2:–** Roteta et al. (2019) claim that burned area estimates in Africa are more than 80% higher when using 20m Sentinel-2 data compared to MCD64A1. Their result is incompatible with the results of this work, so there needs to be a more nuanced explanation of what uncertainty is in the context of this study given that comparison of two products at 250m resolution is not the same as one product at 250m and another at 20m.
**AC:** We note that we do not make comparisons of products at 250m/500m but instead at a lower resolution (1 degree). This point is important because the TC method described here is not suitable at the pixel scale where burned area is a categorical variable (although approximate methods have been developed e.g.McColl et al., 2016). As such the comparison of the two 250m products and a 20m product is the same within the TC framework – because the products are aggregated to a shared 1 degree grid. The comparison against the 20m Sentinel-2 would indicate that coarse resolution products have a natural bias towards under-estimation – which is a systematic error and can not be estimated from the TC method here. We would therefore refer to the

text added to Section 5 detailed above. We also note in the text added to Section 5 above that the Sentinel-2 burned area product is not a validation dataset – and should not be treated as such.

**RC2:** A more important issue is the recent advances in burned area product validation which were not accounted for in this manuscript. For example, in P2 L9-11, the (uncited) claim is made that "Even the largest and most sophisticated validation datasets correspond to only a small sampling of global fire activity, and it is not clear whether this is sufficient information to build an understanding of uncertainties at global and decadal scales". The work done by Boschetti et al., 2016; Padilla et al., 2015; and Padilla et al., 2017 show that uncertainty can, in fact, be identified at the global scale using stratified random sampling.

**AC:** We would suggest that it is difficult or even impossible for even these large-scale validation activities to provide unique and well-characterised uncertainties globally because of the difficulties of scaling. The referenced papers provide omission/commission statistics averaged over many pixels for selected sites. Scaling from these statistics to global spatio-temporally dense uncertainties is not straightforward. This is for example what is done to provide standard uncertainties for the GFED product: $\sigma_B^2 = c_B A$ The uncertainty coefficients $c_B$ are estimated against validation data and applied as a multiplicative error on burned area $A$ to provide a standard uncertainty. For GFED this involves three unique values of $c_B$ globally. While larger validation datasets could effectively further refine $c_B$ it is not obvious how to spread the point estimates from the validation data into $c_B$ or a similar statistical measurement uncertainty model. Parameterising the statistical model would naturally involve an interpolation process of the spatio-temporally sparse validation statistics. For example would the global omission/commission statistic be used or would the per-validation site statistics be interpolated by land cover or geographic region? Ultimately these would still require an uncertainty model similar to the GFED model which is very similar to the multiplicative TC measurement model with the added limitation of the requirement for interpolating from the sparse validation points. To clarify this point we have added the

following to the discussion of the paper (section 6):

"While new large scale validation datasets of burned area have been recently developed (Chuvieco et al., 2018; Padilla et al., 2017), these provide regional-to-global commission/omission error statistics which need to be interpolated with a statistical model of the measurement process to provide explicit spatiotemporally dense uncertainties (such as is done in GFED4). Specifying and then parameterising spatially and temporally such models is a considerable challenge."

**RC1:** On P1 L17, it is stated (again uncited) that the "true information content of such datasets is still unquantified," yet the uncertainties of the FireCCI50 and MCD64A1 products are provided as part of the accuracy assessment done by Chuvieco et al. (2018) as well as within the FireCCI50 gridded product itself (P4 L15). The references to other works which call for the availability of uncertainty data (e.g. Mouillot et al., 2014; Rabin et al., 2017; Yue et al., 2014; Knorr et al., 2014) pre-date the work which has been done through more current validation exercises such as that in Chuvieco et al. (2018) and Roteta et al. (2019).

**AC:** We think that this issue is dealt with in section 2.2 and the reply above. We would repeat from above that however large the validation dataset is, uncertainties for spatio-temporal grid-scale estimates would still need to be extrapolated with a statistical model similar to the in the GFED methodology and described in section 2.2 (pg 4 L3). To clarify this point we have made reference to the validations done by Chuvieco et al. (2018) and Roteta et al. (2019) in the text added to the manuscript above.

**RC2:** Finally, the manuscript needs more context to explain why this work is necessary, especially given that there are Stage 3 burned area validation datasets which can provide estimates of uncertainty in the burned area measurements. The CEOS LPV guidelines for validation stages are referenced in the Discussion, and while the validation stage definitions are somewhat vague for stage 3 (https://lpvs.gsfc.nasa.gov/), "uncertainty" has typically been understood to refer to accuracy with an associated uncertainty accompanying the accuracy estimate. While the TC method in this manuscript

can provide uncertainty estimates, perhaps even compatible with Stage 4, the uncertainty is presented independent of the accuracy of the data set – a user cannot use the uncertainty information alone to know how likely a given pixel is to be correctly labeled. This is of less concern to the modelling community, but should be addressed nonetheless.

**AC:** We suggest that we have covered these points in responses above, for the most part. We have argued above that the validation datasets provide one potential route to uncertainties in burned area but these have specific limitations. In particular issues of representativity and the ability to formulate a statistical model to extrapolate these point estimates to a global temporally and spatially dense quantification of product uncertainties. The TC method presented provides an alternative method – which as we make clear could provide a useful companion to the omission/commission statistics provided by validation datasets.

We agree that users should be made aware of how accurate products are the pixel level ("how likely a given pixel is to be correctly labeled") and would recommend algorithms move towards pixel level uncertainties. The TC estimates can not directly provide per-pixel uncertainties but neither can validation datasets for pixels outside of the validation. We would suggest that pixel QA information is probably the closest to this (with the exception of FireCCI50 pixel-level uncertainties). To address these issues we have added comment: "While the TC-estimated uncertainties can not directly provide information on uncertainties at the pixel level, we would also encourage users to consider the quality assurance (QA) information provided in these products". The reviewer states that "uncertainty has typically been understood to refer to accuracy with an associated uncertainty accompanying the accuracy estimate" but this does not correspond to definitions of uncertainty used across several fields which utilise these data products (i.e. dynamic vegetation models, climate change users & emission users/modellers of these products etc.) The IPPCC Guidelines provide a useful definition of uncertainty for these communities: "Lack of knowledge of the true value of a variable that can be described as a probability density function (PDF) characterising the range and likelihood of possible values. Uncertainty depends on the analyst's state of knowledge, which in turn depends on the quality and quantity of applicable data as well as knowledge of underlying processes and inference methods." (https://www.ipcc-nggip.iges.or.jp/public/2006gl/pdf/1_Volume1/V1_3_Ch3_Uncertainties.pdf) To make this clearer in the text we have added the following relevant definition of uncertainty to the introduction (section 1):

"The trust that users can place into these products can be improved by providing estimates of product uncertainty. This entails providing a quantitative statement about the lack of knowledge of the true burned area described by a probability density function (PDF) characterising the range and likelihood of possible values (ISO/BPIM, 2008; IPCC 2006)."

**Specific Comments**

**RC2:** The abstract needs to be rewritten. The first sentence ends with a dangling participle ("these datasets" refers to nothing); the study period should be included in the abstract; the sentence about the uncertainty estimates is unclear – at minimum it should note that the estimates are per year, but the phrase "Theoretical uncertainties indicate constraints. . ." is unnecessarily complicated given that the values are simply burned area estimates with uncertainty; "product" should be "products"; why are Africa and Australia singled out in the abstract?

**AC:** We have changed "mean global burned area" to "mean annual global burned area" to clarify that these are annual estimates. We have fixed the typo for "products" and referred to the study period. We have rephrased the sentence beginning "Theoretical uncertainties indicate constraints. . ." Africa and Australia are singled out because the three products show the majority of burned area in these continents. The new abstract reads: "Quantitative information on the error properties of global satellite-derived burned area (BA) products is essential for evaluating the quality of these products e.g. against modelled BA estimates. We estimate theoretical uncertainties for three widely-used global satellite-derived BA products using a multiplicative triple collocation error

model. The approach provides spatially-unique uncertainties at 1° for the MODIS Collection 6 burned area product (MCD64); the MODIS Collection 5.1 MCD45 product and the FireCCI50 product for 2001-2013. The uncertainties on mean global burned area for three products are NUM for MCD64, NUM for FireCCI50, and NUM for MCD45. These correspond to relative uncertainties of 4–5.5% and also indicate previous uncertainty estimates to be underestimated. Relative uncertainties are 8–10% in Africa and Australia for example and larger in regions with less annual burned area. The method provides uncertainties that are likely to be more consistent with modelling and data analysis studies due to their spatially explicit properties. These properties are also intended to allow spatially explicit validation of current burned area products."

**RC2:** A definition of uncertainty should be provided to distinguish between uncertainty in total burned area vs (for example) temporal uncertainty in day of burning. This is also important in light of the findings of Roteta et al. (2019) whose results are incompatible with these using the conventional understanding of uncertainty.
**AC:** We think is made clear by the clarification of the uncertainty definition added to the introduction section detailed above.

**RC2:** P2 L7 "validation exercises": Some of the previously referenced studies are intercomparisons, not validation exercises - the former does not imply accuracy assessment. For example, in Humber et al. (2018), no assumption is made that any one product is correct and in fact it is possible that all four products are incorrect for any given burn.
**AC:** We agree and we have rephrased this section to make it clear that Humber et al. (20198) is an intercomparison study.

**RC2:** P3 L22-23 "Simon et al. [. . .]": This does not need to be included, all algorithms have parameters which lead to commission/omission errors.
**AC:** We would prefer to leave this in as we think it is important to provide some broader context on this to readers not acquainted with the limitations of remote sensing/burned area mapping algorithms.

**RC2:** P7: How is the aggregation affected by temporal uncertainty, such as that indicated by the MCD64A1 SDS or the nominal 8-day uncertainty of the MCD45A1 product? In theory even a 1-day shift in burn date detection could lead a cell to be excluded erroneously from a 16-day period and the temporal uncertainty of MCD45A1 is in fact greater than half of the compositing period. Generally, given that this is a paper about uncertainty, it would be good to incorporate the temporal uncertainty in the measurements somehow.

**AC:** We would suggest that the nominal 8-day uncertainty of the MCD45A1 product is as a nominal value a theoretical overestimate. Figure 7 in Giglio et al. (2018) details a good agreement in the detection date between MCD64 and MCD45. An 8-day disagreement for MCD45 against MCD64 occurs in around 2-3% of the pixels considered. Practically this is also not possible to implement into the TC method. We have highlighted this issue as a potential limitation of our approach in the manuscript (pg 17, L9).

**RC2:** P8: Generally, it would benefit the reader to have a discussion of the fire seasonality – the calendar year has been shown to be a fairly bad cutoff period for burned area. In Figure 3, the reader would benefit from knowing that the peaks in uncertainty correspond to the peak of the burning season in Australia. The legend for the figure should also indicate that this figure refers to Australia, and the figure on the right is missing the x-axis labels.

**AC:** We thank the reviewer for the suggestions about figure 3 and have clarified this in the manuscript with "Large absolute uncertainties are associated with the peak in the burning season [...]". We have also improved the caption for the figure.

**RC2:** P12 L8: The problem with the definition of uncertainty is very evident here ("[. . .] cropland burning with relative uncertainties of 8-10%.") The products generally agree with each other about cropland burning, however they are all severely underestimating the total amount (See Hall et al., 2016 which demonstrated MCD45A1 and MCD64A1 underestimate agricultural burning by > 90%). This illustrates that the uncertainty presented here is relative to the other products, underscoring the need for ground data as a baseline for comparison

**AC:** We do not argue with the reviewer that ground data is needed for validating products and note that the TC method is designed to complement validation in the text (pg 19 L10). The reviewer is correct – if the products do agree about the magnitude of cropland burning the random errors are small – the products provide a precise estimate. Of course this does not mean that this magnitude is correct as systematic errors/biases reduce the accuracy of the product (but not it's precision). As before this is a feature that the TC method can only address random errors (eg precision) but not systematic errors (eg. bias). We already discuss sources of uncertainties in croplands in the discussion (Pg 20 L33) and make reference to Hall et al. (2016).

**RC2:** P13: The comparison the MCD64A1 C5.1 might not be relevant, many things about the product were changed and there is not a flat 26% increase in burned area globally – some regions increased significantly more than others, and the detection rate of small fires is significantly higher in the Collection 6 product. Perhaps a better test of the TC method would be to replace the Collection 6 product with Collection 5.1 for the purpose of comparing the result.

**AC:** We considered this but decided that the keep clarity with the rest of the paper – and the use of MCD64C6 as one of the three products in the TC method – it was best to compare the uncertainties directly. Further, because the TC method utilizes all three datasets, changing one will change the estimated uncertainties which makes the comparison across the paper less meaningful. Because of the version difference we make no references to differences in burned area detected between the versions but instead just the relative uncertainties between GFED estimates and the TC estimates.

**RC2:** P19 L5-7 "the majority of current burned area products have only achieved stage two validation": What are the current burned area products referred to in this sentence? Of publicly available operational (global) products, three come to mind (FireCCI51, MCD64A1, Copernicus Burnt Area), and of those two were validated at Stage 3 in

Chuvieco et al. (2018). The reference to Padilla et al. (2014) is actually a strategy for Stage 3 validation (not Stage 2), and the reference should be expanded to include Boschetti et al. (2016) and Padilla et al. (2017), both of which improve upon the temporal robustness of the sample necessary to provide accuracy and uncertainty estimates through time.

**AC:** The reviewer is correct we meant to refer to stage four validation. The text has now been amended: "This study has estimated theoretical uncertainties for three global satellite-derived burned area datasets. This study provides an update on ongoing efforts to provide quantitative uncertainties for remotely sensed global burned area estimates initiated with GFED4 (Giglio et al., 2006b) and continued within the FireCCI products (Chuvieco et al., 2018). Within the four-stage validation scheme developed for land remote sensing products developed by CEOS Land Product Validation (LPV), the majority of current burned area products have only achieved stage threetwo validation (Boschetti et al., 2009; Morisette et al., 2006; Chuvieco et al., 2018; Boschetti et al., 2016; Padilla et al., 2017). Meeting the stage four three requirement for statistically robust and validated uncertainties remains an open challenge for the burned area community."

**RC2:** P20 L17-18: It seems the results are relevant to modelers at coarse resolutions. How would a user implement this work at finer scales?

**AC:** The reviewer is right to suggest that the results are relevant to modellers at coarse resolution. This is because the uncertainty characterisation presented here is carried post hoc on the products at coarse resolution. It is not therefore straightforward (without many assumptions) to downscale these estimates back to the pixel resolution. The correct route to uncertainty at the pixel to grid cell is via uncertainty quantification at the pixel scale as has been prototyped in the FireCCI50 algorithm which is then upscaled to the coarse resolution. As demonstrated in section 4.1.2 these require more work to be consistent but represent a good first step to multi-resolution uncertainties.

**RC2:** P21: How should a user implement the information from this work? What about

reconciling the differences with works like Roteta, who indicated burned area totals in Africa are well outside of the bounds of uncertainty presented in this work?

**AC:** We agree that more discussion about the use of these uncertainties is warranted in the paper. We have therefore added a section to the discussion to describe some prescient uses of the uncertainties: "While the TC-estimated uncertainties can not directly provide information on uncertainties at the pixel level, we would also encourage users to consider the quality assurance (QA) information provided in these products. The presented TC uncertainties have many uses. The uncertainties could, for example, be used to drive development and refinement 35 of parameters in dynamic vegetation models related to fire processes or improve optimisation routines for parameter selection (Poulter et al., 2015; Forkel et al., 2019). They could also be used to better constrain uncertainties on emission estimates derived from 'bottom-up' inventory approaches (Randerson et al., 2012; French et al., 2004; Knorr et al., 2012; Van Der Werf et al., 2017). Explicit uncertainties additionally allow for the development of more advanced assimilation of the satellite observations into models through mathematical frameworks in data assimilation." We have addressed the differences between systematic error and random errors above and added the text discussing these to Section 5.

**Technical Comments:**

**RC2:** P5 L30: Should refer to eq. 2-4.
**AC:** corrected.

**RC2:** P6 Figure 1: Typo in legend of figure on the left.
**AC:** thanks.

**RC2:** P7 L1: Specify 1 degree at the equator.
**AC:** we have this to "with a resolution of 1 degree at the equator"

**RC2:** P11 L5: "product's" should be "products"
**AC:** changed.

**RC2:** P13 L32-33: The last sentence is a fragment.
**AC:** This has been fixed.

**RC2:** P19 L4: The reference to Giglio et al., 2006b
should be with "GFED4"
**AC:** fixed.

**RC2:** P19 L17: Remove parenthesis around Zhu et al.
**AC:** thanks.

**RC2:** – P20 L16-17: References should be in parenthesis.
**AC:** corrected.